materials science

shape memory, hydrogen bond elastomer, toughness, polyurethane

**Author for correspondence:**
Wei Gao
e-mail: galaxy@gxu.edu.cn

This article has been edited by the Royal Society of Chemistry, including the commissioning, peer review process and editorial aspects up to the point of acceptance.

# The effect of promoting hydrogen bond aggregation based on PEMTC on the mechanical properties and shape memory function of polyurethane elastomers

Muqun Wang[1], Shaofeng Liang[1], Wei Gao[1,2] and Yuxuan Qin[1]

[1]School of Resources, Environment and Materials, Guangxi University, Nanning 530000, Guangxi, People's Republic of China
[2]Guangxi Engineering and Technology Research Center for High Quality Structural Panels from Biomass Wastes, Nanning 530000, Guangxi, People's Republic of China

In this work, small molecule diols named PEMTC were synthesized from isophorone diisocyanate, N-(2-hydroxyethyl) acrylamide and trimethylolpropane by a semi-directional method. PEMTC (2-(prop-2-enamido)ethyl N-{3-[({[2-ethyl-3-hydroxy-2(hydroxymethyl)propoxy]carbonyl}amino)methyl]-3, 5,5-trimethylcyclohexyl}carbamate) contains hydrogen bond active site and light-initiated C=C. We introduced it as a branch chain block into poly(ε-caprolactone) (PCL). By feeding and monitoring the reaction process, we synthesized a large number of polyurethane elastomers, hydrogen bonds PCL-based elastomer (HPE), which contain a large number of dynamic hydrogen bonds. Under UV irradiation, PEMTC can make HPE molecules aggregate and cross-link, improve the degree of internal hydrogen bonding interaction of HPE materials and endow HPE materials with good elasticity, toughness, heat resistance and shape memory ability. After 270 nm UV irradiation, the elongation at break of HPE materials decreased from 607.14–1463.95% to 426.60–610.36%, but the strength at break of HPE materials increased from 3.36–13.52 to 10.28–41.52 MPa, and the toughness increased from 16.36–129.71 to 40.48–172.22 MJ m$^{-3}$. In addition, the highest shape fixation rate of HPE after UV irradiation was 98.0%, and the recovery rate was 93.7%.

# 1. Introduction

Shape memory polymers are a kind of material that can give one or more temporary shapes and can be returned to the initial state by external field stimulation, such as heat [1,2], electric [3,4], solvent [5], magnetic field [6,7], humidity [8,9], light [10,11] and pH [12,13]. It has a wide application prospect in biomedical [14,15], self-repairing [16,17], intelligent textile [18,19], drug controlled release [20,21], aerospace [22,23] and other fields. Shape memory polyurethane (SMPU) has attracted the attention of researchers because of its advantages of structural design diversity, easy processing and good biocompatibility. SMPU can be regarded as a block copolymer of a soft segment and a carbamate-based hard segment [24,25]. There are many hydrogen bonds between hard segments, which bind the movement of hard segments and make the hard segments tend to aggregate, while the soft segments can deform greatly [26–29]. Although the compatibility between the two chains is poor, it can be connected by a chemical bond, which makes polyurethane form a microphase separation structure [30,31]. The thermal response of SMPU can be understood as follows: when the temperature is higher than the glass transition temperature ($T_g$) of the soft segment or the melting temperature ($T_m$) of the crystallization zone, the soft segment with a high elastic state will undergo large deformation, while the hard segment with glass state will prevent the molecular chain from sliding and produce internal resilience; when it is cooled to low temperature, the deformation will be fixed; when it is heated again above $T_g$ or $T_m$ of the soft segment, the hard segment will be deformed. Releasing the stored internal stress restores the material to its original shape [1,2]. It is the synergism between the soft segment and hard segment that makes polyurethane possess a shape memory effect (SME). Therefore, the necessary condition for polyurethane to possess a SME is that the content of the hard segment and the relative molecular weight of the soft segment should be controlled in a proper range. One way to improve the shape memory effect of polyurethane materials is to analyse the functional group hydrogen bond ratio between hard segment molecular chains, improve the hydrogen bond content and conversion and improve the shape memory performance of materials. In previous studies, most of them only consider how to improve the hydrogen bond content and improve the shape memory properties of polyurethane materials by increasing the proportion of hydrogen-bonded segments [32,33]. In order to change this situation, on the basis of previous studies, we synthesized a small-molecular-weight glycol PEMTC (2-(prop-2-enamido)ethyl N-{3-[({[2-ethyl-3-hydroxy-2(hydroxymethyl)propoxy]carbonyl}amino)methyl]-3,5,5-trimethylcyclohexyl}carbamate) which can be polymerized by UV irradiation. PEMTC was introduced into the molecular chain of polyurethane by block polymerization to provide a short branched chain containing C=C for UV irradiation polymerization. When the polyurethane film is cured, the secondary cross-linking is initiated by UV irradiation, which makes a large number of hydrogen bonds get closer, improves the degree of hydrogen bond content in the hard segment and achieves the purpose of improving the SME of the material. This process can effectively improve the internal hydrogen bond aggregation of materials, save the amount of chain extender, reduce the raw material cost in the synthesis process and is of great significance to realize low-carbon and environmental protection in the material production process.

# 2. Experimental

## 2.1. Chemicals and materials

Poly(ε-caprolactone) (PCL, Mn = 2000 g mol$^{-1}$), Daicel Chemical Industry Co., Ltd, Japan, decompression dehydration treatment before use; polyethylene glycol (PEG, Mn = 400 g mol$^{-1}$), Shanghai Aladdin Biochemical Technology Co., Ltd, dehydrated under reduced pressure before use; isophorone diisocyanate (IPDI), purity 96%, Shanghai Aladdin Biochemical Technology Co., Ltd, vacuum distillation treatment before use; N-(2-hydroxyethyl)acrylamide (HEMAA), purity 99%, Shanghai Aladdin Biochemical Technology Co., Ltd, dehydrated and dried by activated molecular sieve before use; 2,6-di-tert-butyl-p-cresol (BTH), chemically pure, Shanghai Aladdin Biochemical Technology Co., Ltd, not treated before use; 1,4-butanediol (BDO), purity 99%, Shanghai Aladdin Biochemical Technology Co., Ltd, dehydrated and dried by activated molecular sieve before use; trimethylolpropane (TMP), analytical grade, Shanghai Aladdin Biochemical Technology Co., Ltd, vacuum dried at 90°C before use; pentaerythritol (PER), analytical grade, Shanghai Aladdin Biochemical Technology Co., Ltd, vacuum dried at 90°C before use; dibutyltin dilaurate (DBTDL), purity 99%, Shanghai Aladdin Biochemical Technology Co., Ltd, not treated before use; 2-hydroxy-1-[4-(2-hydroxyethyl)phenyl]-2-methyl-1-propiophenone (rgacure2959), purity 99%,

**Figure 1.** The diagram of synthesis of PEMTC.

Shanghai Aladdin Biochemical Technology Co., Ltd, not treated before use; N,N-dimethylformamide (DMF), purity 99%, Shanghai Aladdin Biochemical Technology Co., Ltd, not treated before use; Acetone (Ac), analytical grade, Sino pharm Chemical Reagent Co., Ltd; not treated before use.

## 2.2. Preparation of PEMTC

Using IPDI and HEMAA as raw materials, through a mild urethanization reaction, to react to produce 2-(prop-2-enamido)ethyl N-[3-(isocyanate methyl)-3,5,5-tris methylcyclohexyl]carbamate (PEITC). PEITC reacts with TMP to form 2-(prop-2-enamido)ethyl N-{3-[({[2-ethyl-3-hydroxy-2-(hydroxymethyl) propoxy]carbonyl}amino)methyl]-3,5,5-trimethylcyclohexyl}carbamate (PEMTC). Figure 1 shows the synthesis path of PEMTC.

The synthesis process is carried out in two steps. In the first step, add equimolar IPDI and HEMAA into a three-necked flask. The flask was equipped with a thermometer, a magnetic stirrer and $N_2$ protection device. Then added an appropriate amount of DBTDL as a catalyst and keep the reaction in a cold water bath at 20°C. -NCO (isocyanate, R-N=C=0) content is measured, the reaction is stopped when -NCO content reaches the theoretical value and PEITC is obtained. In the second step, add excess TMP to the three-necked flask and add an appropriate amount of BHT (2,6-di-tert-butyl-4-methylphenol) as a polymerization inhibitor, dissolve it with acetone, dissolve the product PEITC obtained in the first step into acetone and add it dropwise to the three-necked flask, keep the temperature at 50°C and react until the product, the -NCO absorption peak disappeared in the infrared spectrum. An appropriate amount of acetone can be added to reduce the viscosity during the reaction. Finally, the product is poured into a large amount of deionized water to dissolve the excess TMP and acetone in the deionized water. After filtration, a viscous solid is obtained, which is placed in an oven at 60°C and dried under a vacuum. After removing the water, after separation and purification, the final product PEMTC is obtained. The content of -NCO was measured by di-n-butyl amine-hydrochloric acid titration [34,35]. For these titrations, put a stirrer and 1 g sample into a conical flask, dissolve the sample with 25 ml dibutyl amine toluene solution and react. After the sample was stirred at room temperature for 20–30 minutes, isopropanol was added for dilution, and a few drops of bromocresol green were added for colour reaction. Standard solution of hydrochloric acid (0.1 mol l$^{-1}$) was used for titration to change the solution from blue to yellow and the amount of hydrochloric acid was recorded. The -NCO content was calculated according to the following equation:

$$\text{-NCO\%} = \frac{42c \times (V_1 - V_2)}{1000m} \times 100\%, \tag{2.1}$$

where -NCO% is the content of isocyanate, i.e. percentage content; $V_1$ and $V_2$ are the volumes of hydrochloric acid standard titration solution consumed by blank test and sample respectively (ml); $c$ and $m$ are the concentration of standard hydrochloric acid titration solution (mol l$^{-1}$) and the mass of sample, respectively (g). At the same time, the intermediates of different reaction times in the reaction process were analysed by Fourier transform infrared spectrometry (FT-IR), the area of -NCO absorption peak was calculated by integration and the titration results were compared to improve the data accuracy, to achieve accurate control of the reaction process.

## 2.3. Preparation of HPEs

Change the molar ratio of PEMTC/BDO/PEG400/TMP/PER, set up the orthogonal experiment to synthesize a series of hydrogen bonds PCL-based elastomers (HPE) with different hydrogen bonds content and molecular weight. HPEs are denoted as HPE-0 to HPE-10, respectively; the sample formula is shown in electronic supplementary material, table S1.

**Figure 2.** The diagram of synthesis of HPEs.

In a 500 ml flask with agitator, thermometer, condensation reflux device and nitrogen protection device, add PCL and IPDI in a metered ratio, and at the same time, add an appropriate amount of DBTDL as a catalyst; the temperature is slowly raised to 90°C under the protection of N₂. During the reaction, the content of -NCO in the system is measured, and when the theoretical value is reached, the temperature of the system is reduced to 40°C to obtain prepolymer A. After the synthesis reaction temperature is cooled to 40°C, metered amounts of PEMTC, BDO, PEG400, TMP, PER and appropriate amounts of BTH and Ac are added step by step to reduce the viscosity of the system and then slowly increase the temperature to 80°C. When the -NCO content theoretical value is reached, prepolymer B is obtained. Cool prepolymer B to 40°C, add the appropriate amount of HEMAA and Ac and then raise the temperature to 60°C to react until -NCO was completely consumed. The system is reduced to room temperature. Finally, the solvent Ac is removed by vacuum distillation to obtain a uniform and stable polyurethane liquid. The crude product was washed three times with petroleum ether (3 × 100 ml), and unreacted drugs were removed by centrifugation. Figure 2 shows the synthesis path of HPEs.

Add 3 wt% photoinitiator Irgacure 2959 to HPE liquid stir well. Then, pour it into a polytetrafluoroethylene mould and put it in an oven, vacuum dry at 50°C to remove moisture until the weight does not change. After the sample is dried and cured, irradiate it with a 270 nm UV lamp for 10 min.

## 2.4. Characterization

Mechanical properties tests were carried out using a 6800 electronic universal material tensile testing machine (INSTRON, USA) equipped with 500 N load cells to test its mechanical properties. FT-IR uses Nicolet iS 50 FT-IR (Thermo Fisher, Inc.) with the attenuated total reflection mode for real-time monitoring of samples during the reaction. Nuclear magnetic resonance spectroscopy (¹H NMR spectroscopy) was performed on AVANCE III HD500 nuclear magnetic resonance spectrometer (Bruck, Germany). Use chloroform or dimethyl sulfoxide as the solvent to dissolve the sample. With tetramethylsilane as the internal reference, ¹H NMR spectra were obtained. Wide-angle X-ray diffraction (WAXD) patterns were recorded using XRD-6100 (Rigaku D/MAX 2500 V, Rigaku

Corporation) with Cu K$\alpha$ radiation. The data were collected between 5° and 60° with a scanning speed of 10°C min$^{-1}$. Morphology characterization dissolves the sample in DMF, prepares a sample solution with a mass concentration of 10 wt% and casts it on a copper grid. The agent will slowly evaporate at room temperature. The product will be transferred to a vacuum oven at 40°C for 24 h to remove residual solvents. Use HT77OO TEM (Hitachi, Japan) to observe the degree of microscopic phase separation, and its acceleration voltage is 120 kV. Thermal measurement thermogravimetric analysis used DTG-60(H) (Shimadzu Corporation, Japan) to study the thermal degradation behaviour of samples; the heating rate is 10°C min$^{-1}$. The SME measurement is carried out on a dynamic mechanical thermal analyser (DMA 850, TA Company, USA) with a frequency of 1 Hz [36,37]. Firstly, the temperature was reduced to 10°C for 3 min and then the temperature was increased to $T_m + 5$°C at the rate of 5 K min$^{-1}$. Constant stress was applied after 5 min of isothermal treatment (the corresponding stress value when the sample reached 100% strain was selected according to the stress–strain curve of the sample at 60°C). When the strain reaches 100%, remove the stress quickly, and a temporary strain ($\varepsilon_{\mathrm{load}}$) is obtained. At this time, the sample will have a certain rebound. When the spring-back stops and the deformation of the sample is fixed, the strain ($\varepsilon_f$) is obtained. At last, raise the temperature rapidly to $T_m + 5$°C. When the deformation of the sample is unchanged, the final strain ($\varepsilon_r$) is obtained. Finally, the shape fixity ratio ($R_f$) and shape recovery ratio ($R_r$) were calculated using the following equations (3.1) and (3.2), where $\varepsilon_0$ is the initial length:

$$R_f = \frac{\varepsilon_f - \varepsilon_0}{\varepsilon_{\mathrm{load}} - \varepsilon_0} \times 100\% \tag{3.1}$$

and

$$R_r = \frac{\varepsilon_f - \varepsilon_r}{\varepsilon_f - \varepsilon_0} \times 100\%. \tag{3.2}$$

# 3. Results and discussion

## 3.1. Synthesis and characterization of PEMTC

IPDI is an alicyclic diisocyanate containing two -NCO groups with different reactivity. Because the primary -NCO group in the IPDI molecule is hindered by the cyclohexane ring and the α-substituted methyl group, the secondary -NCO group attached to the cyclohexane is more reactive than the primary -NCO group. In the case of DBTDL as a catalyst, the secondary -NCO group directly connected to the alicyclic ring is more than 10 times more reactive than the primary -NCO; the secondary -NCO connected to the alicyclic ring will react preferentially [38,39]. During the synthesis of PEMTC and HPE, under the condition of a slight excess of IPDI, as the synthesis reaction proceeds, the -NCO content in the system will gradually decrease as the reaction proceeds, and the secondary -NCO in IPDI is mainly consumed. We calculated the -NCO in the system by chemical titration and Gauss–Lorentz curve fitting of the real-time infrared spectra content to detect the progress of the reaction, to realize the directional synthesis of the product.

Electronic supplementary material, figure S1 shows the change of -NCO content during the reaction. By adjusting the reaction conditions, a higher purity end -NCO-terminated PEITC can be obtained. The end -NCO of the generated PEITC molecular chain is the primary -NCO in the original IPDI, and the reactivity of each -NCO is the same. After that, the -NCO concentration change during the reaction is monitored by the batch feeding method. Obtain high-purity PEMTC and directional chain extension and directional grafted HPE macromolecules. Under the set reaction conditions, during the reaction process, the -NCO content dropped sharply in the first 120 min. This is due to the high concentration of -NCO groups and hydroxyl groups in the system at the beginning of the reaction, and the reaction speed is fast. Afterwards, the reaction rate decreased due to the decrease of -NCO concentration and hydroxyl group concentration. When the reaction time exceeds 270 min, the -NCO content reaches the theoretical value (12.38%) and remains unchanged.

In figure 3a,b, we can see the stretching vibration peaks of methyl and methylene groups ($v_{\mathrm{CH2}}$ and $v_{\mathrm{CH3}}$) at 2855–2955 cm$^{-1}$. At 1540 cm$^{-1}$, we can see the in-plane bending swing vibration peak of -NH in PEMTC. The deformation vibration $\delta_{\mathrm{C-N}}$ to the C–N bond can be seen from 1380 to 1430 cm$^{-1}$. Besides, comparing the two spectra at 1638 cm$^{-1}$, it can be seen the stretching vibration peak $v_{\mathrm{C=C}}$ of the C=C bond in HEMAA and the product PEMTC. At 1410 cm$^{-1}$, it can be seen the strong in-plane swing vibration of the C–H bond in the =CH$_2$ and =C–H bonds $\delta_{\mathrm{=CH}}$; at 985 cm$^{-1}$, the out-of-plane swing

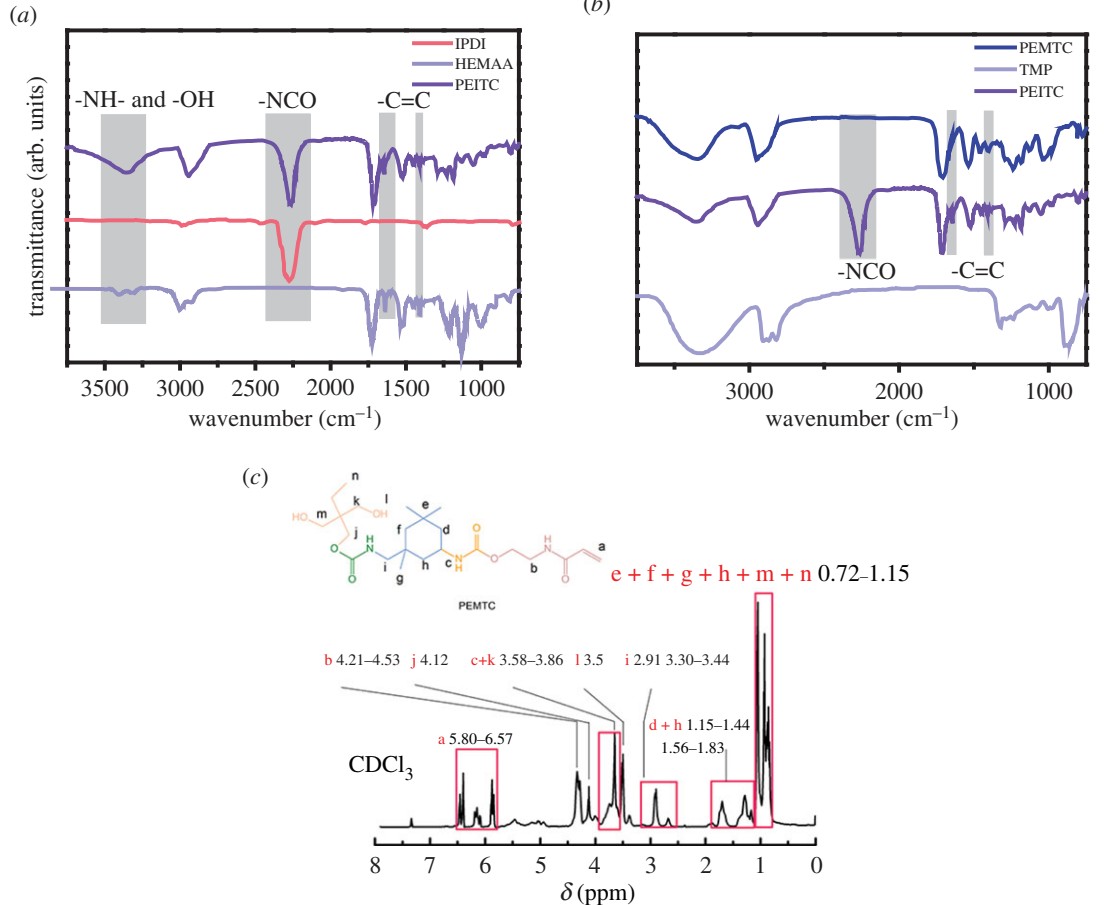

**Figure 3.** FT-IR spectra of PEITC and its reactants (*a*); FT-IR spectra of PEMTC and its reactants (*b*); ¹H NMR spectrum of PEMTC (*c*).

vibration peak of the C–H bond in the $=CH_2$ and $=C$–H bonds can be seen $\delta_{=CH}$. At 810 cm⁻¹, the bending vibration of the C–H bond in the $=C$–H bond can be seen. From this, it can be determined that the acrylic double bond has been successfully incorporated into the intermediate product PEITC and the final product PEMTC. Comparing the infrared spectra of PEITC and PEMTC, two differences can be found. First, in the infrared spectroscopy of PEITC, an obvious characteristic absorption peak of -NCO can be seen at 2267 cm⁻¹, while in the final product PEMTC, this characteristic absorption peak disappears, indicating that the reaction is complete. Second, in the PEITC infrared spectrum, 3346 cm⁻¹ is the stretching vibration peak of -NH in the carbamate. At 3340 cm⁻¹, the PEMTC infrared spectrum peak is wider; this absorption peak corresponds to the stretching vibration peak of N-H in the urethane $v_{-NH}$ and the stretching vibration peak of the hydroxyl group $v_{-OH}$. The generation of PEMTC proved that the photocurable C=C double bond in HEMAA was successfully introduced.

In figure 3*c* and table 1, the ¹H NMR chemical shift map during the reaction confirmed the successful synthesis of the small molecule glycol PEMTC. In theory, HPE synthesized with PEMTC chain extension will have photocurable properties and will provide a large amount of amide group -NH- as a dynamic hydrogen bond donor and lactone bond C=O in PCL to form dynamic hydrogen bond micro-crosslinking regions.

## 3.2. Synthesis and characterization of HPEs

According to the above reaction, characteristics and principles, we used IPDI and PEMTC to carry out semi-directional chain extension polymerization of PCL and set different ratios for each reaction. The structure of PEG and PCL polymerization unit is similar, but different from PCL; PEG polymerization unit does not contain lactone bond C=O, so adding a different proportion of PEG in the reaction process can adjust the arrangement density of hydrogen bond receptors. BDO, PEG, TMP and PER are polyols with the same or similar structural units; they can adjust the hydrogen bond arrangement

**Table 1.** The assignments of $^{1}$H NMR spectrum of PEMTC.

| position of proton | chemical shift (ppm) | theoretical number of proton | integral results of the peaks |
|---|---|---|---|
| a | 5.80–6.57 | 3 | 2.72 |
| b | 4.21–4.53 | 4 | 3.92 |
| j | 4.12 | 2 | 2.03 |
| c + k | 3.58–3.86 | 5 | 4.97 |
| l | 3.5 | 2 | 2.29 |
| i | 2.91 | 2 | 1.97 |
| d + h | 1.15–1.44, 1.56–1.83 | 4 | 3.97 |
| e + f+g + h+m + n | 0.72–1.15 | 18 | 16.72 |

of polyurethane materials according to different branch positions, so as to adjust the crystallinity of HPE materials (electronic supplementary material, figure S2).

We can first synthesize linear macromolecular samples HPE-1, HPE-2, HPE-3 and HPE-4 in the process of directional chain extension and select the experimental group HPE with the best performance HPE-3. Based on the HPE-3 test ratio, BDO was replaced with TMP and PER by sequencing and quantification to obtain samples HPE-5 and HPE-6. TMP and PER, as polyols with different degrees of branching, are used as chain extenders to branch HPE linear macromolecules, and because the degree of branching varies locally, they can form different three-dimensional cross-linked networks. Samples HPE-5 and HPE-6 were obtained. Next, we selected a sample HPE-6 with better film-forming properties, and based on the experimental reaction of HPE-6, we further replaced BDO with PEMTC and introduced a curing functional group C=C and a dynamic hydrogen bond donor. By designing different proportions, samples HPE-7, HPE-8 and HPE-9 are obtained.

Figure 4$a$ shows the FT-IR spectra of HPE-6 before and after UV irradiation, and figure 4$b$ is an enlarged image of the key characteristic peak. It is worth noting that before UV irradiation, the stretching vibration peak of C=C bond $v_{C=C}$ can be seen at 1638 cm$^{-1}$, and the strong in-plane rocking vibration of C–H bond of =CH$_2$ and =C–H bond $\delta_{=CH}$ can be seen at 1410 cm$^{-1}$. At 985 cm$^{-1}$, the out-of-plane rocking vibration peaks of C–H bond in =CH$_2$ and =C–H bond $\delta_{=CH}$ can be seen; at 810 cm$^{-1}$, the bending vibration of C–H bond in =C–H bond can be seen; these are the characteristic absorption peaks of C=C bond, so it can be determined that the diols containing double bond are successfully incorporated into the main chain of polyurethane. After curing, the characteristic absorption peak of the double bond weakened or disappeared, which proved that the cross-linking reaction of acrylic acid C=C bond did take place. Before curing, PEMTC was in free state without cross-linking, there is no stable dynamic hydrogen bonding region. By comparing the infrared spectra of HPE-6 before and after curing and cross-linking, we found that the stretching vibration peak $v_{-NH}$ of -NH- bond formed near 3330 cm$^{-1}$ is obviously weakened. This is because hydrogen bonding can increase the bond length of C=O, -NH- and other chemical bonds. The stretching vibration of chemical bonds is inversely proportional to the square root of the bond length, so the wavenumber will decrease.

The hydrogen bond can change the vibration frequency of the chemical bond in the molecule. The peaks between 1800 and 1600 cm$^{-1}$ can be attributed to different C=O stretch bands [40]. When HPEs are irradiated by UV, the C=O stretching peak shifts to a lower wavenumber, and the bandwidth and intensity are enhanced. The peaks were further analysed by differentiating and peak fitting through Gauss–Lorentz curves. Figure 4$c,d$ and electronic supplementary material, figure S3 are FT-IR spectra of all HPEs films fitted by Gauss–Lorentz curves. The black curve is pristine absorption of HPEs films; the light green is the curve fitted by Gauss–Lorentz; the blue, green, red and purple are the absorption of free hydrogen-bonded carbonyl, disordered hydrogen-bonded carbonyl, ordered hydrogen-bonded carbonyl and carbonyl group in crystalline region [41]. Compared with before UV irradiation, we can clearly see that the peak frequency bands of disordered hydrogen-bonded carbonyl groups in HPEs are narrowed and weakened, while the frequency bands of ordered hydrogen-bonded carbonyl groups are broadened and strengthened, and crystalline hydrogen-bonded carbonyl groups are newly formed. There are indications that C=O in HPEs further associates with NH to form more hydrogen bonds after UV irradiation.

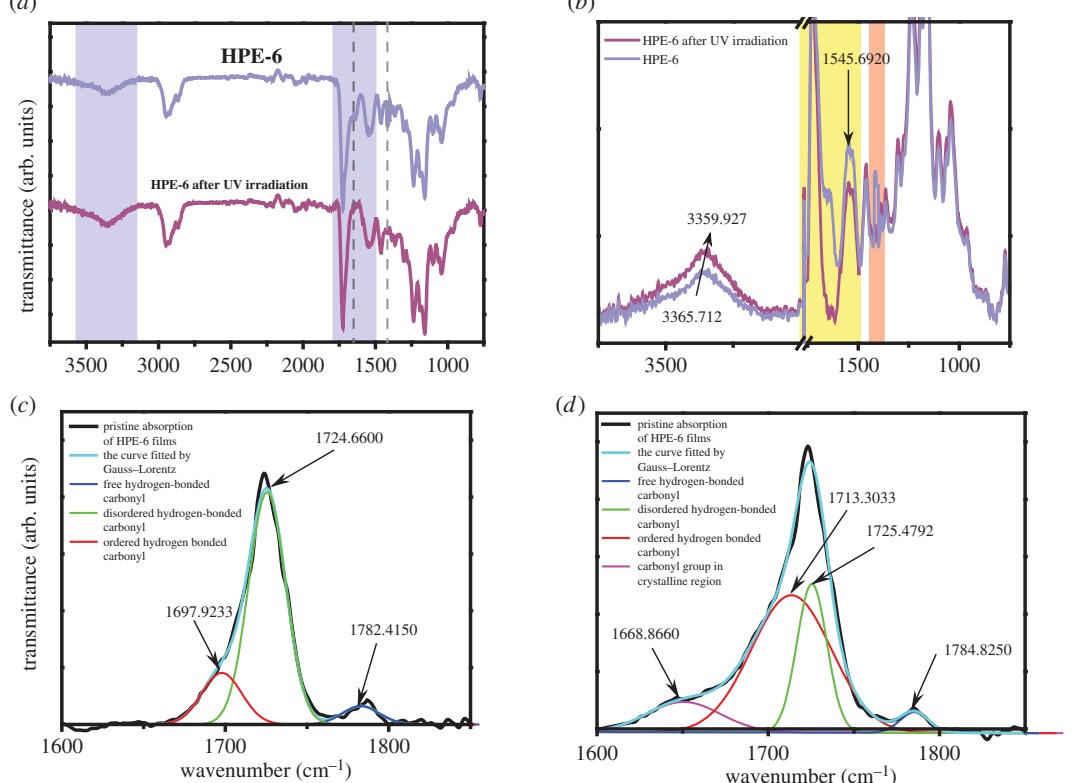

**Figure 4.** FT-IR spectra of HPE-6 before and after UV irradiation (*a*); FT-IR spectra of HPEs (*b*); FT-IR spectra of HPE-6 film before UV irradiation fitted by Gauss–Lorentz curves (*c*) and FT-IR spectra of HPE-6 film after UV irradiation fitted by Gauss–Lorentz curves (*d*).

According to the design of different phases of micro cross-linking in HPE synthesis, we selected the samples HPE-3, HPE-6, HPE-8 and HPE-9 after secondary chain extension for comparison. Infrared analysis of the reactants showed that the characteristic peaks of HPE-3, HPE-6, HPE-8 and HPE-9 were similar. In electronic supplementary material, figure S3(a), with the increase of PEMTC content in the four groups of samples, the stretching vibration peak $v_{N-H}$ of N–H bond near 3380 cm$^{-1}$, the C–N stretching is observed at 1415–1420 cm$^{-1}$, and the stretching vibration $v_{C=O}$ of C=O group at 1720 cm$^{-1}$ decreases obviously. We infer that this is due to the increase of dynamic hydrogen bond content with the increase of PEMTC content.

## 3.3. Mechanical property of HPEs

The formation of hydrogen bonds can improve the tensile strength of HPE material, so we carried out the mechanical tensile test on samples HPE-0, HPE-3, HPE-6, HPE-8 and HPE-9. Figure 5 shows the typical stress–strain curves of HPE materials with different proportions before and after UV irradiation promotes hydrogen bond aggregation.

Before UV irradiation, C=C at the end of PEMTC did not undergo radical polymerization, resulting in cross-linking. According to the set synthesis ratio, the amount of branched-chain extender TMP and PER increased, the amount of straight-chain extender BDO and PEG decreased and the branching degree of HPE increased, the strain of HPE decreased and the tensile strength increased. However, with the increase of the amount of PEMTC in HPE with the same degree of branching, the maximum elongation and tensile strength of the material continue to increase. The stress–strain curves of all HPE samples were always in a tensile yield state before UV irradiation. According to these results, we speculate that this is due to the limited degree of polymerization provided by the hard segment of HPE with a low degree of branching, and the uncross-linked PEMTC can further enhance the molecular length of the branched segment, but it also makes some short straight chains on the HPE molecular chain, which affects the formation of stable crystallization region, so it is difficult for the material to produce elastic deformation in the tensile process. During the tensile yield process, the internal chain slippage and bunching of HPE occur constantly, and internal energy is generated,

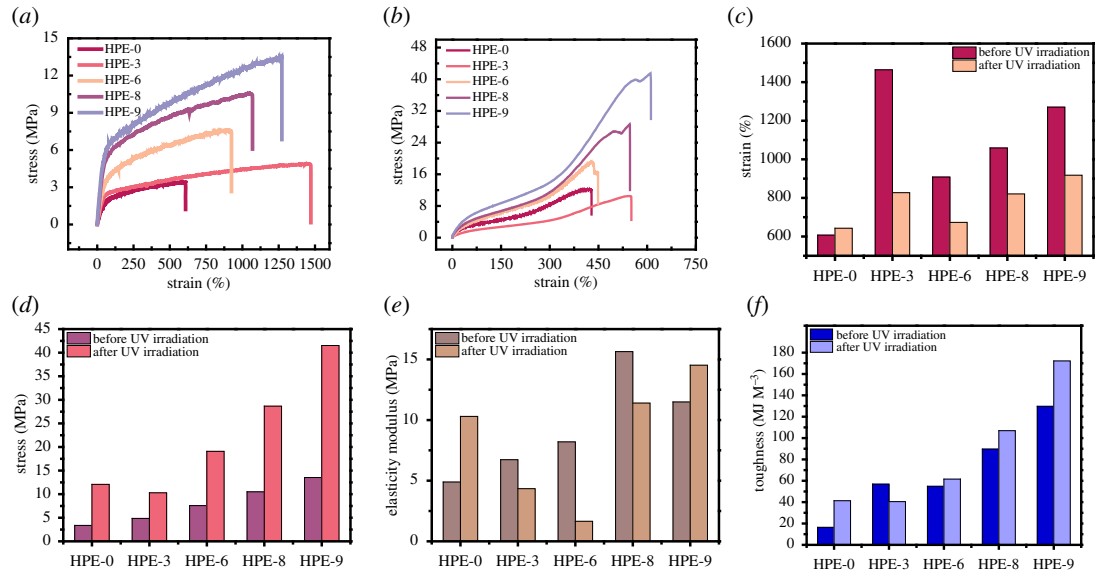

**Figure 5.** Stress–strain curves of HPEs before UV irradiation (*a*); stress–strain curves of HPEs after UV irradiation (*b*); the elongation at break of HPEs before and after UV irradiation (*c*); the break strength of HPEs before and after UV irradiation (*d*); the elasticity modulus of HPEs before and after UV irradiation (*e*) and the toughness of HPEs before and after UV irradiation (*f*).

which makes the N–H group on the branched segment of PEMTC and the C=O group on PCL or other PEMTC form hydrogen bonds. During the tensile process, hydrogen bonds are constantly destroyed and formed, which makes the material present viscoelasticity and improves the tensile strength. After UV irradiation, the typical stress–strain curves of HPE show the situation of yielding first, then strengthening and then yielding. Compared with before UV irradiation, the maximum tensile deformation of HPE decreases, but the tensile strength increases significantly. It can be seen from the above that ultraviolet radiation can generate free radical bonding between the C=C at the end of PEMTC in HPEs molecules, thereby increasing the hydrogen bond content in HPE and enhancing the cross-linking strength of the material. It is worth noting that the tensile strength of HPE-0 after UV irradiation is higher than that of HPE-3, which is because the proportion of PEMTC in HPE-0 is higher than that of HPE-3, thus forming a closer cross-linking network. After HPEs are irradiated with ultraviolet rays, the -C=C in PEMTC will combine with free radicals, and the molecular chains will be cross-linked; this makes it easier to combine N–H and -C=O, form hydrogen bonding interaction, enhances the crystallinity of the material and endows the material with resilience. At the beginning of the stretching process, lattice remodelling occurs, which shows tensile yield and then enters the elastic deformation stage. After the elastic deformation reaches the limit, the tensile yield occurs again, which is similar to that at the end of the stretching before UV irradiation and enters the viscoelastic stage.

## 3.4. Crystal structure of HPEs

According to the [1]H NMR spectra of the samples, we notice that HPE molecules do not have regular structure and symmetry, and it is difficult to crystallize due to molecular rearrangement. Moreover, the introduction of two polyols branched-chain extenders, TMP and PER, further prevents the formation of HPE molecular crystalline regions, so HPE molecules cannot form a large number of microcrystalline regions independently. To further verify the rationality of the test results, we carried out small-angle X-ray scattering and WAXD detection on five groups of samples HPE-0, HPE-3, HPE-6, HPE-8 and HPE-9 after UV irradiation. The WAXD curve is shown in figure 6. All the samples show the diffraction peak characteristics of PCL crystal at $2\theta = 21.39°$, $22.00°$ and $23.70°$ corresponding to (110), (111) and (200) reflections of the orthorhombic cell at A = 7.47 Å, B = 4.98 Å and C = 17.05 Å [42,43].

It can be seen from figure 6 that the intensity of the PCL crystal peak of HPE-3 is greatly weakened compared with HPE-0. This is due to the reduction in the amount of PEMTC added during the synthesis of HPE-3 and the addition of linear chain extenders BDO and PEG, which reduces the degree of

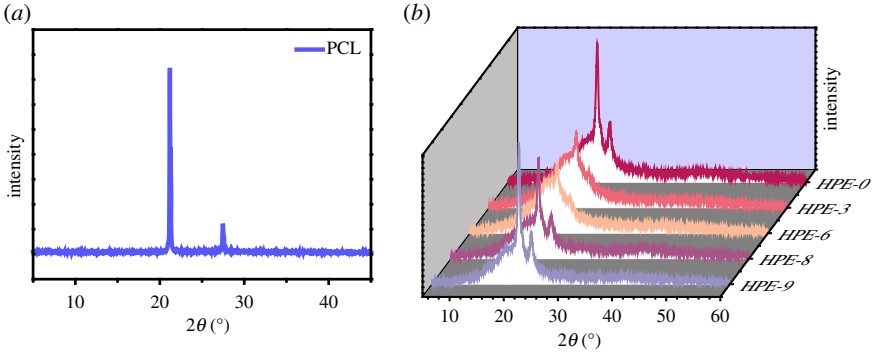

**Figure 6.** Wide-angle X-ray diffraction (WAXD) curves of PCL (*a*); WAXD curves of HPE-0, HPE-3, HPE-6, HPE-8 and HPE-9 (*b*).

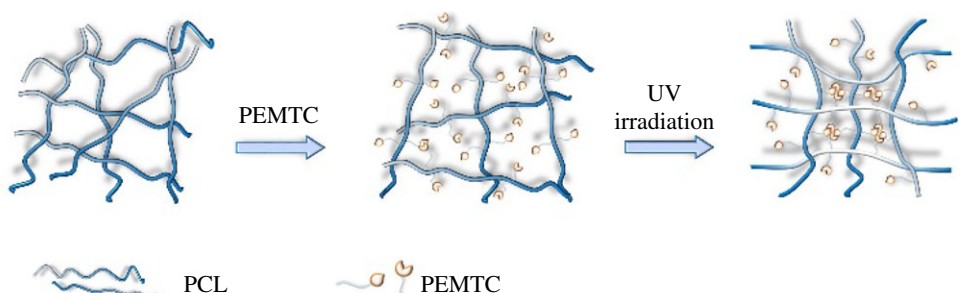

**Figure 7.** Schematic diagram of polymerization principle.

cross-linking and order of the molecular chain after curing. Correspondingly, we can find that comparing the three samples of HPE-6, HPE-8 and HPE-9, the PCL crystallization peak increases with the increase of PEMTC content. The content of PEMTC in HPE-6 and HPE-3 is the same, but the crystallization peak strength of HPE-6 is lower than that of HPE-3 due to the existence of TMP and PER.

Figure 7 is a schematic diagram of the polymerization principle of HPEs. PEMTC makes HPE molecules cross-link after UV irradiation and promotes the formation of hydrogen bonds, making the molecular arrangement more orderly. Analysing figure 6, the diffraction curves of all samples include the gentle characteristic peaks in the amorphous region and the sharp characteristic peaks in the crystalline region; this means that the molecular structure of the HPEs sample has a microphase separation. In figure 8, through TEM images, we can see the micro cross-linking of the sample more intuitively. The aggregation degree of HPES micelles can be improved by increasing the amount of PEMTC. The white dotted circle indicates the typical island structure.

## 3.5. Thermal property of HPEs

Hydrogen bond, as a reversible non-covalent intermolecular force, can greatly improve the SME of shape memory materials. Besides, it also has a certain influence on the thermal properties of materials, such as softening temperature ($T_s$) and $T_m$. The $T_s$ and $T_m$ of the sample can be used as the $T_g$ for the shape memory characterization. The samples were detected by thermogravimetric analysis (TGA) and differential thermal analysis (DTA), and the DTG and DTA curves were analysed.

As shown in figure 9, the results show that the main weight loss range of HPE samples after UV irradiation is 300–400°C. With the increase of PEMTC dosage, the peak position of DTG curve, i.e. thermal decomposition temperature, $T_d$, gradually shifts to the right and increases from 350.15°C to 409.16°C. The reason is that the cross-linking strength of HPE molecules is improved after C=C polymerization induced by UV irradiation. The cross-linking strength and heat resistance of the material increase with the increase of the amount of PEMTC. By observing the DTA heat flow curve, it can be found that there are two endothermic zones in each of the five groups of samples. The softening and melting of HPE occur at 47.02–57.95°C. The starting point of this range is close to the softening temperature $T_s$ of the material, and the peak value is close to the melting temperature $T_m$ of the material. It is worth noting that HPE-0 has better thermal stability than HPE-9. This is due to the addition of a certain amount of BDO and PEG to HPE-9, which reduces the ratio of PEMTC, and the

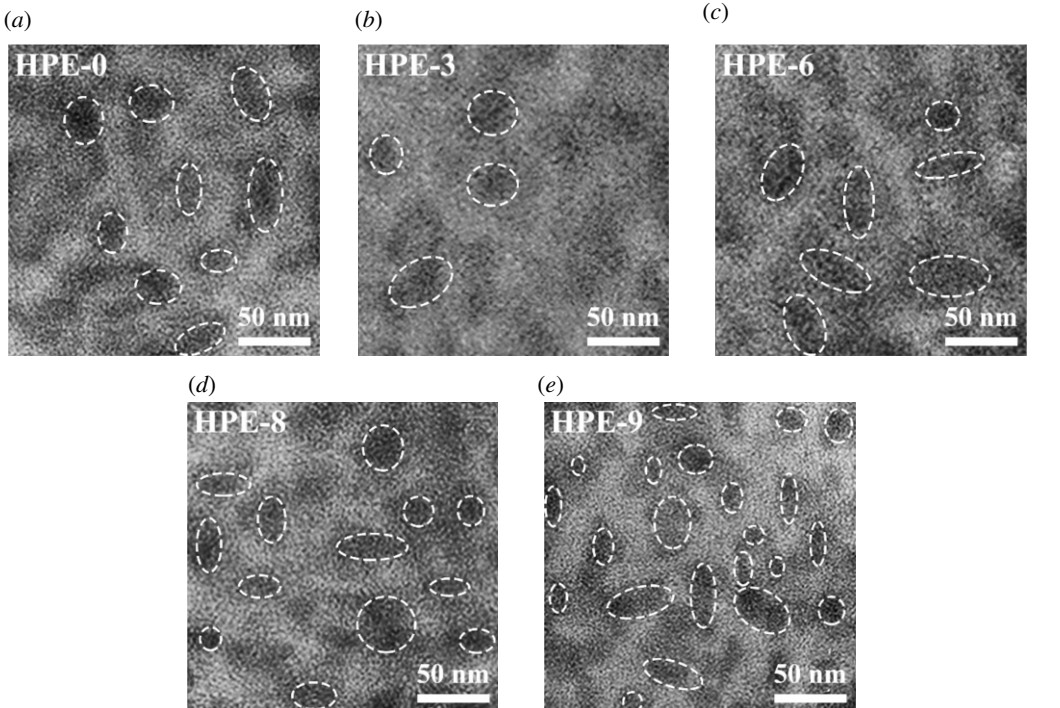

**Figure 8.** TEM images of samples HPE-0 (*a*), HPE-3 (*b*), HPE-6 (*c*), HPE- 8 (*d*) and HPE- 9 (*e*); the scale bar is 50 nm.

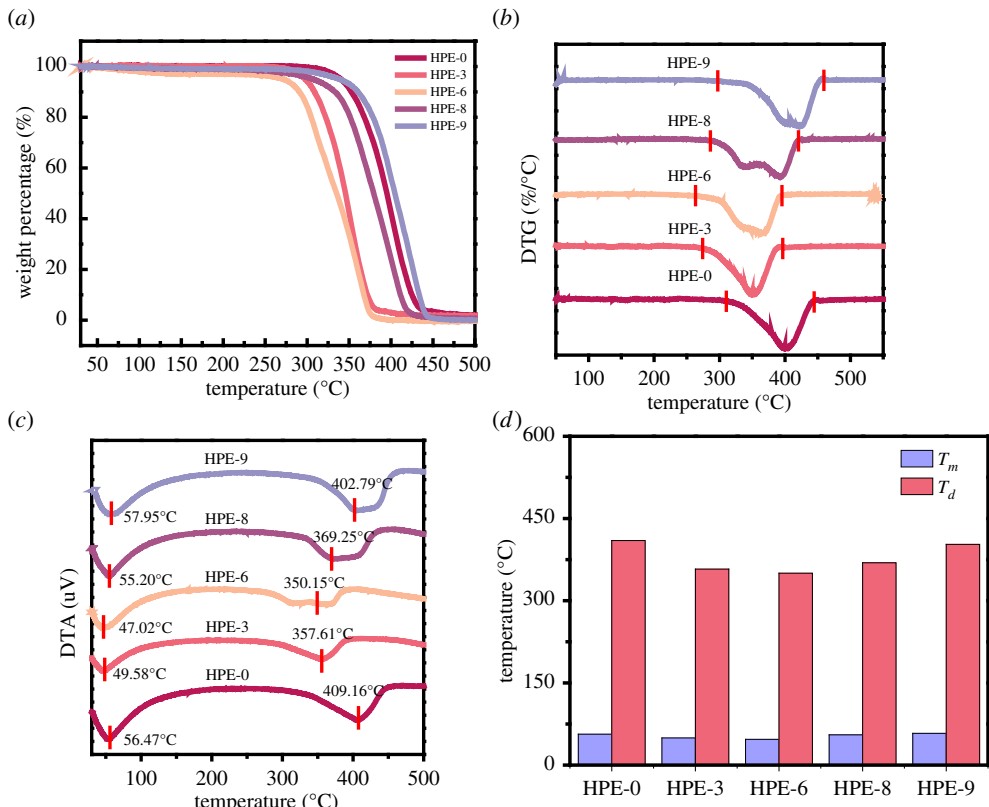

**Figure 9.** TGA curves of HPEs (*a*); DTG curves of HPEs (*b*); DTA curves of HPEs (*c*) and melting temperature and decomposition temperature of HPEs (*d*).

degree of cross-linking is lower than HPE-0. The expansion of endothermic peak area in the range of softening melting temperature and thermal decomposition temperature indicate that PEMTC can provide chemical and dynamic hydrogen bonding double cross-linking, promote the crystallization of

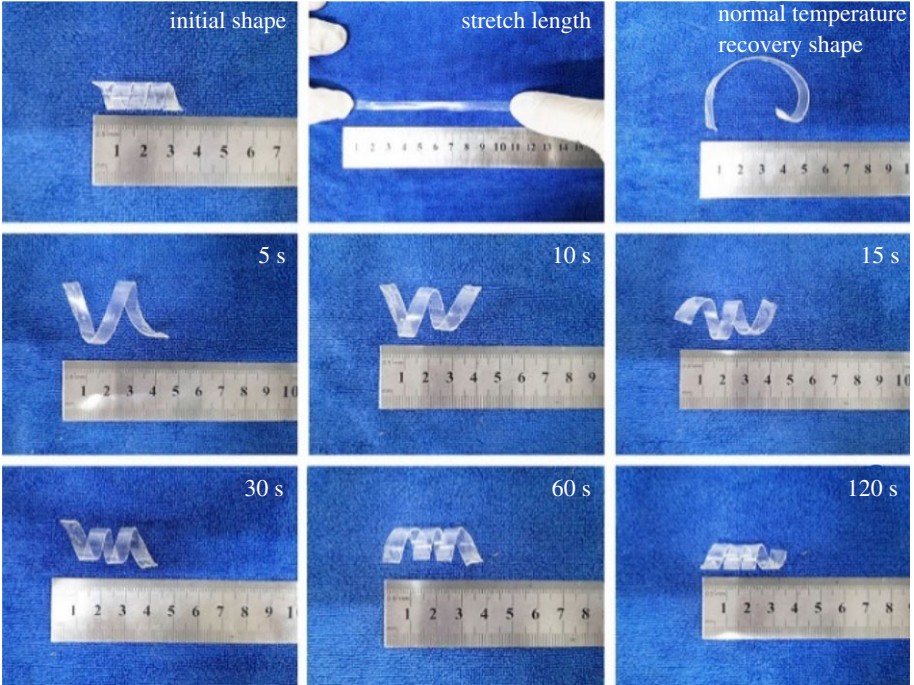

**Figure 10.** Shape memory behaviour of HPE-6 at 60°C.

sample molecules and improve the heat resistance of the sample, which is also consistent with the above analysis.

## 3.6. Shape memory properties of HPEs

The sample was cut into $110 \times 8 \times 1$ mm$^3$, heated at 80°C for 30 min to eliminate the heat history, wound on the metal rod and fixed in a spiral shape and placed in the refrigerator at 0°C for 2 h. Then, on the universal mechanical testing machine, under the condition of constant load, 100% deformation was fixed for 3 h, and the room temperature was taken to recover. After recovering to no deformation, it was put on the heating table and heated for 5–120 s at the temperature close to $T_m$ of the sample, and the shape recovery of the sample was observed. Figure 10 shows the recovery of sample HPE-6. It is found that under simple environmental conditions, the spline can be restored to more than 90% of the original shape by heating in the 120 s. When the temperature is 10–20°C higher than the $T_m$ of the sample, it can recover to more than 60% of the fixed amount of shape in 10 s and more than 80% in the 20 s.

The SME of the materials was further characterized by SME. Figure 11*a* shows the stress–strain curve of the sample in the 100% strain range after isothermal treatment at 60°C for 3 min.

According to the DTA images, it can be found that the $T_m$ of HPE samples is about 60°C, so the mechanical properties of all samples decrease greatly after treatment at 60°C for 3 min. However, due to the different degrees of cross-linking, crystallization and other factors, the heat resistance of different samples is different, and the decline range is different. According to the thermal environment tensile test performance of each sample, we determined the shape memory DMA test parameters of the sample.

Figure 11 shows the shape memory DMA curves of HPE-0, HPE-3, HPE-6, HPE-8 and HPE-9. All samples were not pre-stretched before testing. We found that the whole tensile process of all samples can be divided into five stages according to the different temperature and stress conditions [36,37]. In the initial stage, the temperature is maintained at $T_m + 5°C$, and the sample is only affected by its weight. In the second stage, the temperature is stable, and the sample receives a constant tensile force. In the third stage, the tensile force is constant, and the temperature begins to drop. In the fourth stage, the temperature is maintained at a low level and constant, and the tensile force is also constant; in the final stage, the external force is removed, and the temperature rises rapidly. In the first stage, the sample is not stretched by an external force, but the temperature has been maintained at $T_m + 5°C$. After endothermic, the temperature of the sample begins to rise and soften, a certain amount of remoulding occurs in the crystal region, and the molecular chain rearranges, showing thermal

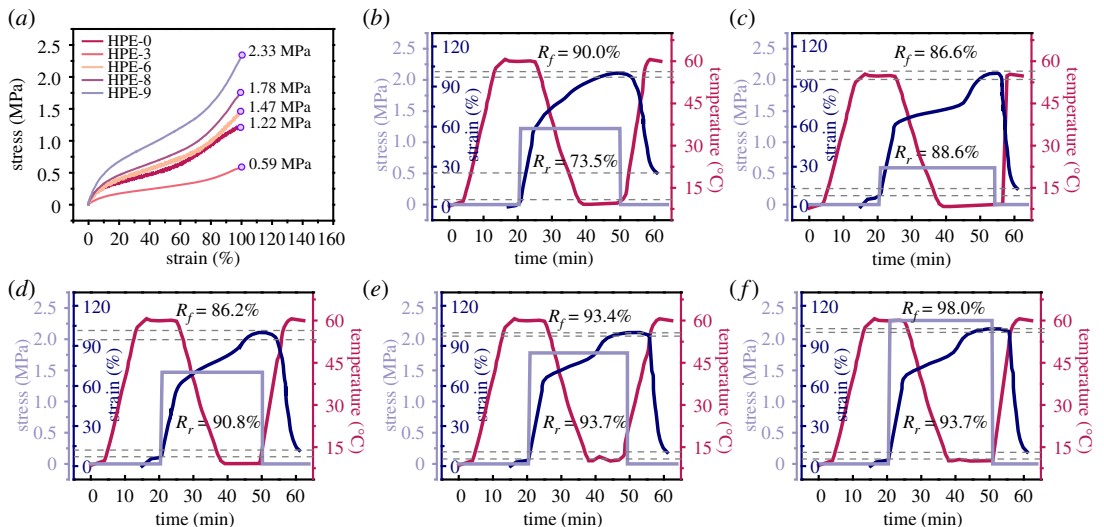

**Figure 11.** Stress–strain curve of HPEs at 60°C (*a*); shape memory DMA curves of HPE-0 (*b*), HPE-3 (*c*), HPE-6 (*d*), HPE-8 (*e*) and HPE-9 (*f*).

expansion, which leads to the formation of deformation under its gravity traction. In the second stage, the temperature remains constant, that is, $T_m + 5°C$ of each sample; the sample softens, presents a high elastic state and begins to bear a constant external tension. At this time, the deformation of the sample is less than 100%, the tensile strength is higher than the resilience provided by the elastic potential energy of the sample, and the intermolecular chain slip occurs. Therefore, the strain rate of the sample shows an obvious upward trend in this stage. In the third stage, the specimen is continuously stretched under constant force, and the temperature begins to drop. After the low two-stage stretching, the molecular chains of the sample are bunched. Under the action of the hard segment stationary phase, the chain slip is limited, and the intermolecular hydrogen bonds recombine freely under the suitable temperature conditions, which further limits the formation of the deformation variables of the sample, so the deformation speed is slowed down. This stage is the main storage stage of sample recovery potential energy. In the fourth stage, the external force keeps constant until it disappears suddenly, and the temperature also drops to 10°C and remains constant. Dynamic hydrogen bonding slows down. Under the influence of intermolecular forces such as hydrogen bond and elastic contraction force caused by the restriction of the hard segment in the molecular chain, the internal stress is generated in the region where crystallization has occurred. The position and shape remain unchanged, the shape is fixed, the deformation recovery energy continues to be stored and finally, the equilibrium is reached. Due to the weak intermolecular force and no hard segment fixation, the amorphous region without crystallization continues to produce chain slip, increasing deformation speed. After the external force is removed, the shape of the sample is fixed. In the final stage, the temperature rises rapidly from low temperature to the $T_m$ of the sample. The recovery energy and shape fixed amount of the sample stored by hydrogen bond are released in a short time, and the shape begins to recover rapidly and finally reaches equilibrium. It is worth noting that HPE-0 with the highest PEMTC content has the least obvious differentiation in the third and fourth stages. We speculate that the high crystallinity of HPE-0 due to the short molecular chain, a high proportion of hard segment stationary phase, and uniform hydrogen bond aggregation lead to the low-temperature edge reaction of HPE-0. HPE-0 has a low degree of chain extension, a high degree of branching and a short molecular chain, and its $R_r$ value is significantly lower than other samples. It can be seen from the figure that except for HPE-0 samples, $R_f$ and $R_r$ of other samples will increase with the increase of PEMTC content. When the temperature is close to the $T_m$, the deformation can recover rapidly within 5 min. Therefore, the introduction of PEMTC can greatly improve the cross-linking strength, hydrogen bonding degree, shape recovery ability, mechanical properties and heat resistance of HPE materials.

# 4. Conclusion

A series of polyurethane elastomers with shape memory function were synthesized by using PEMTC as a chain extender, and small molecule diol PEMTC was synthesized from IPDI, HEMAA and TMP. It was

found that the addition of PEMTC in the synthesis process of polyurethane can make the cross-linking between polyurethane molecules after 270 nm UV irradiation, a large number of functional groups which can form hydrogen bonds gather, the degree of hydrogen bond polymerization is improved, and the mechanical properties and shape memory function of the elastomer material are enhanced.

Data accessibility. The data that support the findings of this study are openly available in Dryad Digital Repository: https://datadryad.org/stash/share/LwZ5_75X-njnv3cB2hj0qjdfMOAMJX5ui4Sd2kfuPrg [44].

The data are provided in electronic supplementary material [45].

Authors' contributions. M.W.: data curation, methodology, software, writing—original draft; S.L.: data curation, methodology; W.G.: funding acquisition, resources, supervision, validation, writing—review and editing; Y.Q.: investigation, methodology.

All authors gave final approval for publication and agreed to be held accountable for the work performed therein.

Competing interests. There are no conflicts to declare.

Funding. We received no funding for this study.

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
