## [Peer Review File · Royal Society Open Science]

Review History

RSOS-211393.R0 (Original submission)

Review form: Reviewer 1

Is the manuscript scientifically sound in its present form?

Yes

Are the interpretations and conclusions justified by the results?

No

Is the language acceptable?

No

Do you have any ethical concerns with this paper?

No

Have you any concerns about statistical analyses in this paper?

No

Recommendation?

Reject

Comments to the Author(s)

The author synthesized a polyurethane structure modified with carbon-carbon double bonds, and studied the effect of hydrogen bonding on the shape memory effect by means of ultraviolet light-initiated polymerization. However, many statements in the article lack data support, such as how to use XRD to know the change of aggregation degree? How to distinguish the effect of hydrogen bond shape memory rather than the degree of crosslinking? How to prove that hydrogen bonds can increase or decrease the bond length of -C=O and -NH-? No DMA data is used to characterize shape memory performance. And there are many grammatical and input errors. Other issues are highlighted in the attachment (see Appendix A).

Review form: Reviewer 2**Is the manuscript scientifically sound in its present form?**

No

Are the interpretations and conclusions justified by the results?

No

Is the language acceptable?

Yes

Do you have any ethical concerns with this paper?

No

Have you any concerns about statistical analyses in this paper?

Yes

Recommendation?

Major revision is needed (please make suggestions in comments)

Comments to the Author(s)

The manuscript reported a polyurethane elastomers with shape memory function, but the formation of reported elastomers are too complicated and writing should be improved. HRMS of PEITC and PEMTC should be provided. The results of small-angle X-ray scattering should also be provided.

Review form: Reviewer 3**Is the manuscript scientifically sound in its present form?**

Yes

Are the interpretations and conclusions justified by the results?

Yes

Is the language acceptable?

Yes

Do you have any ethical concerns with this paper?

No

Have you any concerns about statistical analyses in this paper?

No

Recommendation?

Accept with minor revision (please list in comments)

Comments to the Author(s)

The review file is attached (see Appendix B).

Decision letter (RSOS-211393.R0)

Dear Miss Qin:

Title: The effect of promoting hydrogen bond aggregation based on PEMTC on the mechanical properties and shape memory function of polyurethane elastomers
Manuscript ID: RSOS-211393

The editor assigned to your manuscript has now received comments from reviewers. We would like you to revise your paper in accordance with the referee and Subject Editor suggestions which can be found below (not including confidential reports to the Editor). Please note this decision does not guarantee eventual acceptance.

Please submit your revised paper before 18-Dec-2021. Please note that the revision deadline will expire at 00.00am on this date. If we do not hear from you within this time then it will be assumed that the paper has been withdrawn. In exceptional circumstances, extensions may be possible if agreed with the Editorial Office in advance. We do not allow multiple rounds of revision so we urge you to make every effort to fully address all of the comments at this stage. If deemed necessary by the Editors, your manuscript will be sent back to one or more of the original reviewers for assessment. If the original reviewers are not available we may invite new reviewers.

When submitting your revised manuscript, you must respond to the comments made by the referees and upload a file "Response to Referees" in "Section 6 - File Upload". Please use this to document how you have responded to the comments, and the adjustments you have made. In

order to expedite the processing of the revised manuscript, please be as specific as possible in your response.

Please also include the following statements alongside the other end statements. As we cannot publish your manuscript without these end statements included, if you feel that a given heading is not relevant to your paper, please nevertheless include the heading and explicitly state that it is not relevant to your work.

- Ethics statement

Please clarify whether you received ethical approval from a local ethics committee to carry out your study. If so please include details of this, including the name of the committee that gave consent in a Research Ethics section after your main text. Please also clarify whether you received informed consent for the participants to participate in the study and state this in your Research Ethics section.

OR

Please clarify whether you obtained the necessary licences and approvals from your institutional animal ethics committee before conducting your research. Please provide details of these licences and approvals in an Animal Ethics section after your main text.

OR

Please clarify whether you obtained the appropriate permissions and licences to conduct the fieldwork detailed in your study. Please provide details of these in your methods section.

- Data accessibility

It is a condition of publication that you make available the data and research materials supporting the results in the article. Datasets should be deposited in an appropriate publicly available repository and details of the associated accession number, link or DOI to the datasets must be included in the Data Accessibility section of the article (<https://royalsocietypublishing.org/rsos/for-authors#question17>). Reference(s) to datasets should also be included in the reference list of the article with DOIs (where available).

Please include a Data Availability section after your main text stating where supporting data are available from, or where they will be made available should your article be accepted for publication.

If you wish to submit your supporting data or code to Dryad (<http://datadryad.org/>), or modify your current submission to dryad, please use the following link:
<http://datadryad.org/submit?journalID=RSOS&manu=RSOS-211393>

- Competing interests

Please include a Competing Interests section after your main text declaring any financial or non-financial competing interests. If you have no competing interests please state 'I/we have no competing interests.'

- Authors' contributions

Please include an Authors' Contributions section at the end of your main text detailing the contribution of each author. All authors should have read and approved the manuscript before submission and this should be stated in the Authors' Contributions section.

The list of Authors should meet all of the following criteria; 1) substantial contributions to conception and design, or acquisition of data, or analysis and interpretation of data; 2) drafting the article or revising it critically for important intellectual content; and 3) final approval of the version to be published.

- Acknowledgements

- Funding statement

Please include a funding section after your main text which lists the source of funding for each author.

Yours sincerely,
Dr Ellis Wilde
Publishing Editor, Journals

On behalf of the Subject Editor Professor Anthony Stace and the Associate Editor Professor Chaohua Cui.

RSC Associate Editor
Comments to the Author:
(There are no comments.)

RSC Subject Editor
Comments to the Author:
(There are no comments.)

Reviewers' Comments to Author:

Reviewer: 1

Comments to the Author(s)

The author synthesized a polyurethane structure modified with carbon-carbon double bonds, and studied the effect of hydrogen bonding on the shape memory effect by means of ultraviolet

light-initiated polymerization. However, many statements in the article lack data support, such as how to use XRD to know the change of aggregation degree? How to distinguish the effect of hydrogen bond shape memory rather than the degree of crosslinking? How to prove that hydrogen bonds can increase or decrease the bond length of -C=O and -NH-? No DMA data is used to characterize shape memory performance. And there are many grammatical and input errors. Other issues are highlighted in the attachment.

Reviewer: 2

Comments to the Author(s)

The manuscript reported a polyurethane elastomers with shape memory function, but the formation of reported elastomers are too complicated and writing should be improved. HRMS of PEITC and PEMTC should be provided. The results of small-angle X-ray scattering should also be provided.

Reviewer: 3

Comments to the Author(s)

The review file is attached.

Author's Response to Decision Letter for (RSOS-211393.R0)

See Appendix C.

RSOS-211393.R1 (Revision)

Review form: Reviewer 1

Is the manuscript scientifically sound in its present form?

Yes

Are the interpretations and conclusions justified by the results?

Yes

Is the language acceptable?

Yes

Do you have any ethical concerns with this paper?

No

Have you any concerns about statistical analyses in this paper?

No

Recommendation?

Accept as is

Comments to the Author(s)

The manuscript can be accepted in its present form.

Review form: Reviewer 2

Is the manuscript scientifically sound in its present form?

Yes

Are the interpretations and conclusions justified by the results?

Yes

Is the language acceptable?

Yes

Do you have any ethical concerns with this paper?

No

Have you any concerns about statistical analyses in this paper?

No

Recommendation?

Accept as is

Comments to the Author(s)

Authors have revised the manuscript. The manuscript can be accepted in now form.

Review form: Reviewer 3

Is the manuscript scientifically sound in its present form?

Yes

Are the interpretations and conclusions justified by the results?

Yes

Is the language acceptable?

Yes

Do you have any ethical concerns with this paper?

No

Have you any concerns about statistical analyses in this paper?

No

Recommendation?

Accept as is

Comments to the Author(s)

I agree to accept the paper as is.

Decision letter (RSOS-211393.R1)

Dear Mr Wang:

Title: The effect of promoting hydrogen bond aggregation based on PEMTC on the mechanical properties and shape memory function of polyurethane elastomers
Manuscript ID: RSOS-211393.R1

It is a pleasure to accept your manuscript in its current form for publication in Royal Society Open Science. The chemistry content of Royal Society Open Science is published in collaboration with the Royal Society of Chemistry.

Yours sincerely,
Dr Ellis Wilde
Publishing Editor, Journals

On behalf of the Subject Editor Professor Anthony Stace and the Associate Editor Professor Chaohua Cui.

RSC Associate Editor
Comments to the Author:
(There are no comments.)

RSC Subject Editor
Comments to the Author:
(There are no comments.)

Reviewer(s)' Comments to Author:

Reviewer: 2

Comments to the Author(s)

Authors have revised the manuscript. The manuscript can be accepted in now form.

Reviewer: 1

Comments to the Author(s)

The manuscript can be accepted in its present form.

Reviewer: 3

Comments to the Author(s)

I agree to accept the paper as is.

Appendix A**ROYAL SOCIETY
OPEN SCIENCE****The effect of promoting hydrogen bond aggregation based
on PEMTC on the mechanical properties and shape memory
function of polyurethane elastomers**

Journal:	Royal Society Open Science
Manuscript ID	RSOS-211393
Article Type:	Research
Date Submitted by the Author:	26-Aug-2021
Complete List of Authors:	Wang, Muqun; Guangxi University Liang, Shaofeng; Guangxi University Gao, Wei; Guangxi University Qin, Yuxuan; Guangxi University,
Subject:	Materials science < CHEMISTRY
Keywords:	shape memory, hydrogen bond elastomer, toughness, polyurethane
Subject Category:	Chemistry

Author-supplied statements

Relevant information will appear here if provided.

Ethics

Does your article include research that required ethical approval or permits?:

This article does not present research with ethical considerations

Statement (if applicable):

CUST_IF_YES_ETHICS :No data available.

Data

It is a condition of publication that data, code and materials supporting your paper are made publicly available. Does your paper present new data?:

Yes

Statement (if applicable):

The data that support the findings of this study are openly available in [Gao, Wei; Wang, Muqun (2021), The effect of promoting hydrogen bond aggregation based on PEMTC on the mechanical properties and shape memory function of polyurethane elastomers, Dryad, Dataset] at https://datadryad.org/stash/share/LwZ5_75X-njnv3cB2hj0qjdfMOAMJX5ui4Sd2kfuPrg

Conflict of interest

I/We declare we have no competing interests

Statement (if applicable):

CUST_STATE_CONFLICT :No data available.

Authors' contributions

This paper has multiple authors and our individual contributions were as below

Statement (if applicable):

Muqun Wang: Conceptualization, Data curation, Formal Analysis, Validation, Writing-original draft, Writing-review&editing. Shaofeng Liang: Writing-review&editing, Software. Wei Gao: Funding acquisition, Resources, Supervision. Yuxuan Qin: Writing-review&editing, Validation

1
2
3
4
5
6
7 **Subject Category:**
8 Chemistry

9 **Subject Areas:**
10 materials science

11 **Keywords:**
12 shape memory, hydrogen bond elastomer, toughness, polyurethane

13 **Author for correspondence:**
14 Wei Gao
15 e-mail: galaxy@gxu.edu.cn

The effect of promoting hydrogen bond aggregation based on PEMTC on the mechanical properties and shape memory function of polyurethane elastomers

Muqun Wang^a, Shaofeng Liang^a, Wei Gao^{ab*}, Yuxuan Qin,^a

Abstract: In this work, small molecule diols named PEMTC were synthesized from IPDI, HEMAA, and TMP by a semi-directional method. PEMTC contains hydrogen bond active site and light initiated C=C. We introduced it as a branch chain block into PCL. By feeding and monitoring the reaction process, we synthesized a large number of polyurethane elastomers, HPE, which contain a large number of dynamic hydrogen bonds. Under UV irradiation, PEMTC can make HPE molecules aggregate and crosslink, improve the degree of internal hydrogen bonding of HPE materials, and endow HPE materials with good elasticity, toughness, heat resistance, and shape memory ability. After 270 nm UV irradiation, the elongation at break of HPE materials decreased from 607.14-1463.95% to 426.60-610.36%, but the break strength of HPE materials increased from 3.36-13.52 MPa to 10.28-11.52 MPa, and the toughness increased from 16.36-129.71 MJ·M⁻³ to 40.48-172.22 MJ·M⁻³. Also, the highest shape fixation rate and recovery rate of HPE after UV irradiation were 98.0% and 93.7% respectively.

Introduction

Shape memory polymer (SMPs) is a kind of material that can give one or more temporary shapes and can be returned to the initial state by external field stimulation, such as heat,^{1, 2} electric,^{3, 4} solvent,⁵ magnetic field,^{6, 7} humidity,^{8, 9} light^{10, 11} and pH^{12, 13}. It has a wide application prospect in biomedical,^{14, 15} self-repairing,^{16, 17} intelligent textile,^{18, 19} drug controlled release,^{20, 21} aerospace,^{22, 23} and other fields. Shape memory polyurethane (SMPUs) has attracted the attention of researchers because of its advantages of structural design diversity, easy processing, and good biocompatibility. Shape memory polyurethane can be regarded as a block copolymer of a soft segment and a carbamate-based hard segment.^{24, 25} There are many hydrogen bonds between hard segments, which bind the movement of hard segments and make the hard segments tend to aggregate, while the soft segments can deform greatly.²⁶⁻²⁹ Although the compatibility between the two chains is poor, it can be connected by a chemical bond, which makes polyurethane form a microphase separation structure.^{30, 31} The thermal response of shape memory polyurethane can be understood as follows: when the temperature is higher than the glass transition temperature (T_g) of the soft segment or the melting temperature (T_m) of the crystallization zone, the soft segment with a high elastic state will undergo large deformation, while the hard segment with glass state will prevent the molecular chain from sliding and produce internal resilience; when it is cooled to low temperature, the deformation will be fixed; when it is heated again above T_g or T_m of the soft segment, the hard segment will be deformed to release the stored internal stress to restore the material to its original shape.^{1, 2} It is the synergism between the soft segment and hard segment that makes polyurethane possesses a shape memory effect. Therefore, the necessary condition for polyurethane to possess a shape memory effect is that the content of the hard segment and the relative molecular weight of the soft segment should be controlled in a proper range. One way to improve the shape memory effect of

^aSchool of Resources, Environment and Materials, Guangxi University, Nanning 530000, Guangxi, China. E-mail: galaxy@gxu.edu.cn;

^bGuangxi Engineering and Technology Research Center for High Quality Structural Panels from Biomass Wastes, Nanning 530000, Guangxi, China.

polyurethane materials is to analyze the hydrogen bonding efficiency between hard segment molecular chains, improve the hydrogen bond content and conversion, to improve the shape memory performance of materials. In previous studies, most of them only consider how to improve the hydrogen bond content, and improve the shape memory properties of polyurethane materials by increasing the proportion of hydrogen-bonded segments.^{32, 33} In order to change this situation, on the basis of previous studies, we synthesized a small molecular glycol which can initiate crosslinking under UV irradiation. PEMTC was introduced into the molecular chain of polyurethane by block polymerization to provide a short branched-chain containing C=C for UV irradiation polymerization. When the polyurethane film is cured, the secondary crosslinking is initiated by UV irradiation, which makes a large number of hydrogen bond association sites aggregate, improves the degree of hydrogen bond polymerization in the hard segment, and achieves the purpose of improving the shape memory effect of the material. This process can effectively improve the internal hydrogen bond aggregation of materials, save the amount of modifier, reduce the raw material cost in the synthesis process, and is of great significance to realize low-carbon and environmental protection in the material production process.

Experimental

Chemicals and materials

Poly(ϵ -caprolactone) (PCL, Mn=2000 g/mol), Daicel Chemical Industry Co., Ltd., Japan, decompression dehydration treatment before use; Polyethylene glycol (PEG, Mn=400 g/mol), Shanghai Aladdin Biochemical Technology Co., Ltd., dehydrated under reduced

pressure before use; Isophorone diisocyanate (IPDI), purity 96%, Shanghai Aladdin Biochemical Technology Co., Ltd., vacuum distillation treatment before use; N-(2-Hydroxyethyl)acrylamide (HEMAA), purity 99%, Shanghai Aladdin Biochemical Technology Co., Ltd., dehydrated and dried by activated molecular sieve before use; 2,6-Di-tert-butyl-p-cresol (BTH), chemically pure, Shanghai Aladdin Biochemical Technology Co., Ltd., not treated before use; 1,4-Butanediol (BDO), purity 99%, Shanghai Aladdin Biochemical Technology Co., Ltd., dehydrated and dried by activated molecular sieve before use; Trimethylolpropane (TMP), analytical grade, Shanghai Aladdin Biochemical Technology Co., Ltd., vacuum dried at 90 degrees Celsius before use; Pentaerythritol (PER), analytical grade, Shanghai Aladdin Biochemical Technology Co., Ltd., vacuum dried at 90 degrees Celsius before use; Dibutyltin dilaurate (DBTDL), purity 99%, Shanghai Aladdin Biochemical Technology Co., Ltd., not treated before use; 2-hydroxy-1-[4-(2-hydroxyethyl)phenyl]-2-methyl-1-propionophenone (rgacure2959), purity 99%, Shanghai Aladdin Biochemical Technology Co., Ltd., not treated before use; N,N-Dimethylformamide (DMF), purity 99%, Shanghai Aladdin Biochemical Technology Co., Ltd., not treated before use; Acetone (Ac), analytical grade, Sino pharm Chemical Reagent Co., Ltd.; not treated before use.

Preparation of PEMTC

Using IPDI and HEMAA as raw materials, through a mild urethanization reaction, 2-(prop-2-enamido)ethyl N-[3-(isocyanate methyl)-3,5,5-tris Methylcyclohexyl]carbamate (PEITC). The resulting product is reacted with TMP to finally obtain 2-(prop-2-enamido)ethyl N-[3-[[[2-ethyl-3-hydroxy-2-(hydroxymethyl)propoxy]-carbonyl]amino]methyl]-3,5,5-trimethylcyclohexyl]carbamate, (PEMTC). Fig. 1 shows the synthesis path of PEMTC.

Fig. 1 The diagram of synthesis of PEMTC.

The synthesis process is carried out in two steps. In the first step, add equimolar amounts of IPDI and HEMAA into a three-necked flask equipped with a thermometer, a magnetic stirrer, and N₂ protection, add an appropriate amount of DBTDL as a catalyst and keep the reaction at 20 °C in a cold water bath. -NCO content is measured, the reaction is stopped when -NCO content reaches the theoretical value, and PEITC is obtained. In the second step, add excess TMP to the three-necked flask and add an appropriate amount of BHT as a polymerization inhibitor, dissolve it with acetone, dissolve the product PEITC obtained in the first step into acetone and add it dropwise to the three-necked flask, keep the temperature at 50 °C and react until the product the -NCO absorption peak disappeared in the infrared spectrum. An appropriate amount of acetone can be added to reduce the viscosity during the reaction. Finally, the product is poured into a large amount of deionized water to dissolve the excess TMP and acetone in the deionized water. After filtration, a viscous solid is obtained, which is placed in an oven at 60 °C and dried under a vacuum. After removing the water, after separation and purification, the final product PEMTC is obtained. The content of -NCO was measured by di-n-butyl amine-hydrochloric acid titration.^{34, 35} For these titrations, put a stirrer and 1 g sample into a conical flask, dissolve the sample with 25 mL dibutyl amine toluene solution and react. Shake to make the liquid in the bottle mix evenly, and mix at room temperature for 20-30 min, add isopropanol, add a few drops of bromocresol green as an indicator, titrate with HCl (0.1 mol/L) standard solution when the solution changes from blue to yellow, it is the endpoint, and do the blank experiment. The -NCO content was calculated according to the following equation:

$$-NCO\% = \frac{42c \times (V_1 - V_2)}{1000m} \times 100\% \quad (1)$$

Where -NCO% is the content of isocyanate, i.e. percentage content; V₁ and V₂ are the volumes of hydrochloric acid standard titration solution consumed by blank test and sample respectively (mL); C and m are the concentration of standard hydrochloric acid titration solution (mol/L) and the mass of sample respectively (g). At the same time, the intermediates of different reaction times in the reaction process were

analysed by FT-IR, the area of -NCO absorption peak was calculated by integration, and the titration results were compared to improve the data accuracy, to achieve accurate control of the reaction process.

Preparation of HPEs

Change the molar ratio of PEMTC/BDO/PEG400/TMP/PER, set up the orthogonal experiment to synthesize a series of hydrogen bonds PCL-based elastomers (HPE) with different hydrogen bonds content and molecular weight. HPEs are denoted as HPE-0 to HPE-10, respectively, the sample formula is shown in Table S1.

In a 500 mL flask with agitator, thermometer, condensation reflux device, and nitrogen protection device, add PCL and IPDI in a metered ratio, and at the same time add an appropriate amount of DBTDL as a catalyst, the temperature is slowly raised to 90 °C under the protection of N₂. During the reaction, the content of -NCO in the system is measured, and when the theoretical value is reached, the temperature of the system is reduced to 40 °C to obtain prepolymer A. After the synthesis reaction temperature is cooled to 40 °C, metered amounts of PEMTC, BDO, PEG400, TMP, PER, and appropriate amounts of BHT are added step by step to reduce the viscosity of the system, and then slowly increase the temperature to 80 °C. When the -NCO content theoretical value is reached, prepolymer B is obtained. Cool prepolymer B to 40 °C, add the appropriate amount of HEMA and Ac, and then raise the temperature to 60 °C to react until -NCO was completely consumed. The system is reduced to room temperature, finally, the solvent Ac is removed by vacuum distillation to obtain a uniform and stable polyurethane liquid. Fig. 2 shows the synthesis path of HPEs.

Fig. 2 The diagram of synthesis of HPEs.

Add photoinitiator Irgacure 2959 to HPE liquid, 3wt%, pour it into a polytetrafluoroethylene mold, place it in an oven at 50 °C, vacuum dry to remove moisture, dry to constant weight, and then use 270 nm UV lamp after curing, the final thickness of the film is between 10 and 300 μm.

Characterization

Mechanical properties tests were carried out using a 6800 electronic universal material tensile testing machine (INSTRON, USA) equipped with 500 N load cells to test its mechanical properties. FT-IR uses Nicolet iS 50 FT-IR (Thermo Fisher, Inc.) with the attenuated total reflection (ATR) mode for real-time monitoring of samples during the reaction. Nuclear magnetic resonance spectroscopy (¹H NMR spectroscopy) was performed on AVANCE III HD500 nuclear magnetic resonance spectrometer (Bruker, Germany). Use chloroform (CDCl₃) or dimethyl sulfoxide (DMSO-d₆) as the solvent to dissolve the sample. With tetramethylsilane (TMS) as the internal reference, ¹H NMR spectra were obtained. Wide-angle X-ray diffraction (WAXD) patterns were recorded using XRD-6100 (Rigaku D/MAX 2500V, Rigaku Corporation) with Cu Kα radiation. The data were collected between 5 and 60° with a scanning speed of 10°/min. Morphology characterization Dissolve the sample in DMF, prepare a sample solution with a mass concentration of 10wt%, and cast it on a copper grid. The agent will slowly evaporate at room temperature. The product will be transferred to a vacuum oven at 40 °C for 24 h to remove residual solvents, Use HT7700 TEM (Hitachi, Japan) to observe the degree of microscopic phase separation, and its acceleration voltage is 120 kv. Thermal measurement thermogravimetric analysis (TG) used DTG-60(H) (Shimadzu Corporation, Japan) to study the thermal degradation behavior of samples, the heating rate is 10 °C/min. The shape memory effect (SME) measurement is carried out on a dynamic mechanical thermal analyzer (DMA 850, TA Company, USA) with a frequency of 1 Hz.^{36, 37} Firstly, the temperature was reduced to 10 °C for 3 min, and then the temperature was increased to T_m+5 °C at the rate of 5 K/min. Constant stress was applied after 5 min of isothermal treatment (the corresponding stress value when the sample reached 100% strain was selected according to the stress-strain curve of the sample at 60 °C). When the strain reaches 100%, remove the stress quickly and a temporary strain (ε_{load}) is obtained. At this time, the sample will have a certain rebound. When the

spring backstops and the deformation of the sample are fixed, the strain (ε_f) is obtained. At last, raise the temperature rapidly to T_m+5 °C. When the deformation of the sample is unchanged, the final strain (ε_r) is obtained. Finally, the shape fixity ratio (R_f) and shape recovery ratio (R_r) was calculated using the following Eqn (2) and (3), ε_0 is the initial length:

$$R_f = \frac{\varepsilon_f - \varepsilon_0}{\varepsilon_{\text{load}} - \varepsilon_0} \times 100\% \quad (2)$$

$$R_r = \frac{\varepsilon_f - \varepsilon_r}{\varepsilon_f - \varepsilon_0} \times 100\% \quad (3)$$

Results and discussion

Synthesis and characterization of PEMTC

IPDI is an alicyclic diisocyanate containing two -NCO groups with different reactivity. Because the primary-NCO group in the IPDI molecule is hindered by the cyclohexane ring and the α -substituted methyl group, the secondary-NCO group attached to the cyclohexane is more reactive than the primary-NCO group. In the case of DBTDL as a catalyst, the secondary-NCO group directly connected to the alicyclic ring is more than 10 times more reactive than the primary-NCO, the secondary-NCO connected to the alicyclic ring will react preferentially.^{38, 39} During the synthesis of PEMTC and HPE, under the condition of a slight excess of IPDI, as the synthesis reaction proceeds, the -NCO content in the system will gradually decrease as the reaction proceeds, and the secondary -NCO in IPDI is mainly consumed. We calculated the -NCO content in the system by chemical titration and Gauss-Lorentz curve fitting of the real-time infrared spectra Content to detect the progress of the reaction, to realize the directional synthesis of the product.

Fig. S1 show the change of -NCO content during the reaction. By adjusting the reaction conditions, a higher purity end-NCO-terminated PEITC can be obtained. The end -NCO of the generated PEITC molecular chain is the primary -NCO in the original IPDI, and the reactivity of each -NCO is the same. After that, the -NCO concentration change during the reaction is monitored by the batch feeding method. Obtain high-purity PEMTC and directional chain extension and directional grafted HPE macromolecules. Under the set reaction conditions, during the reaction process, the -NCO content dropped sharply in the first 120 min. This is due to the high concentration of -NCO groups and hydroxyl groups in the system at the beginning of the reaction, and the reaction speed is fast; afterward, due to the -NCO concentration and hydroxyl groups. When the concentration decreases, the reaction speed decreases. When the reaction time exceeds 270 min, the -NCO content reaches the theoretical value (12.38%) and remains unchanged.

In Fig. 3 (a) and (b), we can see the stretching vibration peaks of methyl and methylene groups (ν_{CH_2} and ν_{CH_3}) at 2855-2955 cm^{-1} . At 1540 cm^{-1} , we can see the in-plane bending swing vibration peak of -NH in PEMTC. The deformation vibration $\delta_{\text{C-N}}$ to the CN bond can be seen from 1380 cm^{-1} to 1430 cm^{-1} . Besides, comparing the two spectra at 1638 cm^{-1} , it can be seen that the stretching vibration peak $\nu_{\text{C=C}}$ of the C=C bond in HEMAA and the product PEMTC. At 1410 cm^{-1} it can be seen that the strong in-plane swing vibration of the CH-bond in the =CH₂ and =CH-bonds $\delta_{\text{=CH}}$ at 985 cm^{-1} , the out-of-plane swing vibration peak of the CH bond in the =CH₂ and =CH bonds can be seen $\delta_{\text{=CH}}$. At 810 cm^{-1} , the bending vibration of the CH bond in the =CH bond can be seen. From this, it can be determined that the acrylic double bond has been successfully incorporated into the intermediate product PEITC and the final product PEMTC. Comparing the infrared spectra of PEITC and PEMTC, two differences can be found. First, in the infrared spectroscopy of PEITC, an obvious characteristic absorption peak of -NCO can be seen at 2267 cm^{-1} , while in the final product PEMTC, this characteristic absorption peak disappears, indicating that the reaction is complete. Second, in the PEITC infrared spectrum, 3346 cm^{-1} is the stretching vibration peak of -NH in the carbamate. At 3340 cm^{-1} the PEMTC infrared spectrum peak is wider, this absorption peak corresponds to the stretching vibration peak of -NH in the urethane $\nu_{\text{-NH}}$ and the stretching vibration peak of the hydroxyl group $\nu_{\text{-OH}}$. The generation of PEMTC proved that the photocurable C=C double bond in HEMAA was successfully introduced.

In Fig. 3 (c) and Table (1), the ¹H NMR chemical shift map during the reaction confirmed the successful synthesis of the small molecule glycol PEMTC. In theory, HPE synthesized with PEMTC chain extension will have photocurable properties, and will provide a large amount of amide group -NH- as a dynamic hydrogen bond donor, and lactone bond C=O in PCL. The formation of dynamic hydrogen bond micro-crosslinking area.

Fig. 3 FT-IR spectra of PEITC and its reactants (a); FT-IR spectra of PEMTC and its reactants (b); ^1H NMR spectrum of PEMTC.

Table 1. The assignments of ^1H -NMR spectrum of PEMTC.

Position of proton	Chemical shift (ppm)	Theoretical number of proton	Integral results of the peaks
a	5.80~6.57	3	2.72
b	4.21~4.53	4	3.92
j	4.12	2	2.03
c+k	3.58~3.86	5	4.97
l	3.5	2	2.29
i	2.91	2	1.97
d+h	1.15~1.44, 1.56~1.83	4	3.97
e+f+g+h+m+n	0.72~1.15	18	16.72

Synthesis and characterization of HPEs

According to the above reaction characteristics and principles, we used IPDI and PEMTC to carry out semi-directional chain extension polymerization of PCL and set different ratios for each reaction. The structure of PEG and PCL polymerization unit is similar, but different from PCL, PEG polymerization unit does not contain lactone bond $\text{C}=\text{O}$, so adding a different proportion of PEG in the reaction process can adjust the arrangement density of hydrogen bond receptors. BDO, PEG, TMP, and PER are polyols with the same or similar structural units, they can adjust the hydrogen bond arrangement of polyurethane materials according to different branch positions, so as to adjust the crystallinity of HPE materials (Fig. S2).

We can first synthesize linear macromolecular samples HPE-1, HPE-2, HPE-3, HPE-4 in the process of directional chain extension, and select the experimental group HPE with the best performance HPE-3. Based on the experimental reaction of HPE-3, BDO was replaced with TMP and PER by sequencing and quantification to obtain samples HPE-5 and HPE-6. TMP and PER with different degrees of branching, are used as chain extenders to branch HPE linear macromolecules, and because the degree of branching varies locally, they can form different three-dimensional cross-linked networks. Samples HPE-5 and HPE-6 were obtained. Next, we selected a sample HPE-6 with better film-forming properties, and based on the experimental reaction of HPE-6, we further replaced BDO with PEMTC, and introduced a curing functional group $\text{C}=\text{C}$ and a dynamic hydrogen bond donor. By designing different proportions, samples HPE-7, HPE-8, HPE-9, and HPE-10 were obtained.

Fig. 4 (a) and (b) show the FT-IR spectra of HPE-6 before and after UV irradiation. It is worth noting that before UV irradiation, the stretching vibration peak of $\text{C}=\text{C}$ bond $\nu_{\text{C}=\text{C}}$ can be seen at 1638 cm^{-1} , and the strong in-plane rocking vibration of $\text{C}-\text{H}$ bond of $=\text{CH}_2$ and $=\text{CH}$ bond δ_{CH} can be seen at 1410 cm^{-1} . At 985 cm^{-1} , the out of plane rocking vibration peaks of $\text{C}-\text{H}$ bond in $=\text{CH}_2$ and $=\text{CH}$ bond δ_{CH} can be seen; at 810 cm^{-1} , the bending vibration of $\text{C}-\text{H}$ bond in $=\text{CH}$ bond can be seen; these are the characteristic absorption peaks of $\text{C}=\text{C}$ bond, so

it can be determined that the diols containing double bond are successfully incorporated into the main chain of polyurethane. After curing, the characteristic absorption peak of the double bond weakened or disappeared, which proved that the cross-linking reaction of acrylic acid C=C bond did take place. Before curing, PEMTC was in Free State without cross-linking, there is no stable dynamic hydrogen bonding region. By comparing the infrared spectra of HPE-6 before and after curing and crosslinking we found that the stretching vibration peak ν_{NH} of -NH-bond formed near 3330 cm^{-1} is obviously weakened. **This is because hydrogen bonding can increase the bond length of C=O, -NH- and other chemical bonds. The stretching vibration of chemical bonds is inversely proportional to the square root of the bond length, so the wave number will decrease.**

Fig. 4 FT-IR spectra of HPE-6 before and after UV irradiation (a); FT-IR spectra of HPEs (b); FTIR spectra of HPE-6 film before UV irradiation fitted by Gauss Lorentz curves (c); FTIR spectra of HPE-6 film after UV irradiation fitted by Gauss Lorentz curves (d).

The hydrogen bond can change the vibration frequency of the chemical bond in the molecule. The peaks between 1800 cm^{-1} and 1600 cm^{-1} can be attributed to different C=O stretch bands.⁴⁰ When HPEs are irradiated by UV, the C=O stretching peak shifts to a lower wave number, and the bandwidth and intensity are enhanced. The peaks were further analyzed by differentiating and peak fitting through Gauss Lorentz curves. Fig. 4 (c), (d) and Fig.S3 are FTIR spectra of all HPEs films fitted by Gauss Lorentz curves. The black curve is pristine absorption of HPEs films, the light green is the curve fitted by Gauss Lorentz. The blue, green, red, purple is the absorption of free hydrogen-bonded carbonyl, disordered hydrogen-bonded carbonyl, ordered-hydrogen bonded carbonyl, and carbonyl group in crystalline region.⁴¹ Compared with before UV irradiation, we can clearly see that the peak frequency bands of disordered hydrogen-bonded carbonyl groups in HPEs are narrowed and weakened, while the frequency bands of ordered hydrogen-bonded carbonyl groups are broadened and strengthened, and crystalline hydrogen-bonded carbonyl groups are newly formed. There are indications that C=O in HPEs further associates with NH to form more hydrogen bonds after UV irradiation.

According to the design of different phases of micro crosslinking in HPE synthesis, we selected the samples HPE-3, HPE-6, HPE-8, and HPE-9 after secondary chain extension for comparison. Infrared analysis of the reactants showed that the characteristic peaks of HPE-3, HPE-6, HPE-8, and HPE-9 were similar. In Fig.S3 (a), with the increase of PEMTC content in the four groups of samples, the stretching vibration peak ν_{NH} of -NH bond near 3380 cm^{-1} , the C-N stretching is observed at 1415 cm^{-1} to 1420 cm^{-1} and the stretching vibration $\nu_{\text{C=O}}$ of C=O group at 1720 cm^{-1} decrease obviously. We infer that this is due to the increase of dynamic hydrogen bond content with the increase of PEMTC content.

Mechanical property of HPEs

The formation of hydrogen bonds can improve the tensile strength of HPE material, so we carried out the mechanical tensile test on samples HPE-0, HPE-3, HPE-6, HPE-8, and HPE-9. Fig. 5 shows the typical stress-strain curves of HPE materials with different proportions before and after UV irradiation promotes hydrogen bond aggregation.

Fig. 5 Stress-strain curves of HPEs before UV irradiation (a); stress-strain curves of HPEs after UV irradiation (b); the elongation at break of HPEs before and after UV irradiation (c); the break strength of HPEs before and after UV irradiation (d); the elasticity modulus of HPEs before and after UV irradiation (e); the toughness of HPEs before and after UV irradiation (f).

Before UV irradiation, C=C at the end of PEMTC did not undergo radical polymerization, resulting in cross-linking. According to the set synthesis ratio, the amount of branched-chain extender TMP and PER increased, the amount of straight-chain extender BDO and PEG decreased, and the branching degree of HPE increased, the strain of HPE decreased and the tensile strength increased. However, with the increase of the amount of PEMTC in HPE with the same degree of branching, the maximum elongation and tensile strength of the material continue to increase. The stress-strain curves of all HPE samples were always in a tensile yield state before UV irradiation. According to these results, we speculate that this is due to the limited degree of polymerization provided by the hard segment of HPE with a low degree of branching, and the uncrosslinked PEMTC can further enhance the molecular length of the branched segment, but it also makes some short straight chains on the HPE molecular chain, which affects the formation of stable crystallization region, so it is difficult for the material to produce elastic deformation in the tensile process. During the tensile yield process, the internal chain slippage and bunching of HPE occur constantly, and internal energy is generated, which makes the -NH group on the branched segment of PEMTC and the C=O Group on PCL or other PEMTC form hydrogen bonds. During the tensile process, hydrogen bonds are constantly destroyed and formed, which makes the material present viscoelasticity and improves the tensile strength. After UV irradiation, the typical stress-strain curves of HPE show the situation of yielding first, then strengthening, and then yielding. Compared with before UV irradiation, the maximum tensile deformation of HPE decreases, but the tensile strength increases significantly. It can be seen from the above that ultraviolet radiation can generate free radical bonding between the C=C at the end of PEMTC in HPEs molecules, thereby increasing the hydrogen bond content in HPE, thereby enhancing the crosslinking strength of the material. It is worth noting that the tensile strength of HPE-0 after UV irradiation is higher than that of HPE-3, which is because the proportion of PEMTC in HPE-0 is higher than that of HPE-3, thus forming a closer crosslinking network. **The crosslinking of PEMTC makes the acceptor of dynamic hydrogen bond combine with the donor, enhances the crystallinity of the material, and endows the material with resilience. At the beginning of the stretching process, lattice remodeling occurs, which shows tensile yield, and then enters the elastic deformation stage. After the elastic deformation reaches the limit, the tensile yield occurs again, which is similar to that at the end of the stretching before UV irradiation and enters the viscoelastic stage.**

Crystal structure of HPEs

According to the ¹H NMR spectra of the samples, we notice that HPE molecules do not have regular structure and symmetry, and it is difficult to crystallize due to molecular rearrangement. Moreover, the introduction of two polyols branched-chain extenders, TMP and PER, further prevents the formation of HPE molecular crystalline regions, so HPE molecules cannot form a large number of microcrystalline regions independently. To further verify the rationality of the test results, we carried out small-angle X-ray scattering and wide-angle X-ray diffraction detection on five groups of samples HPE-0, HPE-3, HPE-6, HPE-8, and HPE-9 after UV irradiation. The WAXD curve is shown in Fig. 6. All the samples show the diffraction peak characteristics of PCL crystal at $2\theta=21.39^\circ$, 22.00° , and 23.70° corresponding to (110), (111), and (200) reflections of the orthorhombic cell at $a=7.47 \text{ \AA}$, $B=4.98 \text{ \AA}$, and $C=17.05 \text{ \AA}$.^{42, 43}

如何从XRD图中看出交联程度的...另外,建议补充一个单PCL的XRD对照图

Fig. 6. Wide-angle X-ray diffraction (WAXD) curves of HPEs

It can be found from the Fig. 6 that the PCL crystal diffraction characteristic peaks of HPE-0 and HPE-9 are the strongest, but the crosslinking degree of HPE-0 is low, the molecular chain of HPE-0 is lower than that of HPE-9, and the proportion of PEMTC is higher, so the crystallinity is the most obvious. With the decrease of PEMTC content, the crystallization peaks of other samples weaken. The content of PEMTC in HPE -6 and HPE -3 is the same, but the crystallization peak strength of HPE-6 is lower than that of HPE-3 due to the existence of branched cross-linking.

Fig. 7 Schematic diagram of polymerization principle.

In Fig. 8, through TEM images, we can see the micro cross-linking of the sample more intuitively. The aggregation degree of HPES micelles can be improved by increasing the amount of PEMTC. The white dotted circle indicates the typical island structure.

Fig. 8 TEM images of samples HPE-0 (a), HPE-3 (b), HPE-6 (c), HPE-8(d), and HPE-9(e); the scale bar is 50 nm.

Thermal property of HPEs

Hydrogen bond, as a reversible noncovalent intermolecular force, can greatly improve the shape memory effect of shape-memory materials. Besides, it also has a certain influence on the thermal properties of materials, such as softening temperature (T_s) and melting

temperature (T_m). The T_s and T_m of the sample can be used as the transition temperature for the shape memory characterization. The samples were detected by TGA and DTA, and the DTG and DTA curves were analyzed.

Fig. 9 TGA curves of HPEs (a); DTG curves of HPEs (b); DTA curves of HPEs (c); Melting temperature and decomposition temperature of HPEs.

As shown in Fig. 9, the results show that the main weight loss range of HPE samples after UV irradiation is 300-400 °C. With the increase of PEMTC dosage, the peak position of DTG curve, i.e. thermal decomposition temperature T_d , gradually shifts to the right, and increases from 350.15 °C to 409.16 °C. The reason is that the cross-linking strength of HPE molecules is improved after C=C polymerization induced by UV irradiation. The cross-linking strength and heat resistance of the material increase with the increase of the amount of PEMTC. By observing the DTA heat flow curve, it can be found that there are two endothermic zones in each of the four groups of samples. The softening and melting of HPE occur at 47.02-57.95 °C. The starting point of this range is close to the softening temperature T_s of the material, and the peak value is close to the melting temperature T_m of the material. The expansion of endothermic peak area in the range of softening melting temperature and thermal decomposition temperature indicates that PEMTC can provide chemical and dynamic hydrogen bonding double cross-linking, promote the crystallization of sample molecules, and improve the heat resistance of the sample, which is also consistent with the above analysis.

Shape memory properties of HPEs

The sample was cut into 110×8×1 mm³, heated at 80 °C for 30 min to eliminate the heat history, wound on the metal rod and fixed in a spiral shape, and placed in the refrigerator at 0 °C for 2 h. Then, on the universal mechanical testing machine, under the condition of constant load, 100% deformation was fixed for 3 h, and the room temperature was taken to recover. After recovering to no deformation, it was put on the heating table and heated for 5-120 s at the temperature close to T_m of the sample, and the shape recovery of the sample was observed. Fig. 10 shows the recovery of sample HPE-6. It is found that under simple environmental conditions, the spline can be restored to more than 90% of the original shape by heating in the 120 s. When the temperature is 10-20 °C higher than the T_m of the sample, it can recover to more than 60% of the fixed amount of shape in 10 s and more than 80% in the 20 s.

Fig. 10 Shape memory behaviour of HPE-6 at 60 °C.

The shape memory effect of the materials was further characterized by SME. Fig. 11 (a) shows the stress-strain curve of the sample in the 100% strain range after isothermal treatment at 60 °C for 3 min.

According to the DTA images, it can be found that the melting temperature T_m of HPE samples is about 60 °C, so the mechanical properties of all samples decrease greatly after treatment at 60 °C for 3min. However, due to the different degrees of cross-linking, crystallization, and other factors, the heat resistance of different samples is different, and the decline range is different. According to the thermal environment tensile test performance of each sample, we determined the shape memory DMA test parameters of the sample.

Fig. 11 Stress-strain curve of HPEs at 60 °C(a); Shape memory DMA curves of HPE-0(b), HPE-3(c), HPE-6(d), HPE-8(e) and HPE-9(f).

Fig. 11 show the shape memory DMA curves of HPE-0, HPE-3, HPE-6, HPE-8, and HPE-9. All samples were not pre-stretched before testing. We found that the whole tensile process of all samples can be divided into five stages according to the different temperature and stress conditions.^{36, 37} In the initial stage, the temperature is maintained at T_m+5 °C, and the sample is only affected by its weight. In the second stage, the temperature is stable, and the sample receives a constant tensile force. In the third stage, the tensile force is constant, and the temperature begins to drop. In the fourth stage, the temperature is maintained at a low level and constant, and the tensile force is also constant; in the final stage, the external force is removed, and the temperature rises rapidly. In the first stage, the sample is not stretched by an external force, but the temperature has been maintained at T_m+5 °C. After endothermic, the temperature of the sample begins to rise and soften, a certain amount of remolding occurs in the crystal region, and the molecular chain rearranges, showing thermal expansion, which leads to the formation of deformation under its gravity traction. In the second stage, the temperature remains constant, that is, T_m+5 °C of each sample, the sample softens, presents a high elastic state, and begins to bear a constant external tension. At this time, the deformation of the sample is less than 100%, the tensile strength is higher than the resilience provided by the elastic potential energy of the sample, and the intermolecular chain slip occurs. Therefore, the strain rate of the sample shows an obvious upward trend in this stage. In

the third stage, the specimen is continuously stretched under constant force, and the temperature begins to drop. After the low two-stage stretching, the molecular chains of the sample are bunched. Under the action of the hard segment stationary phase, the chain slip is limited, and the intermolecular hydrogen bonds recombine freely under the suitable temperature conditions, which further limits the formation of the deformation variables of the sample, so the deformation speed is slowed down. This stage is the main storage stage of sample recovery potential energy. In the fourth stage, the external force keeps constant until it disappears suddenly, and the temperature also drops to 10 °C and remains constant. Dynamic hydrogen bonding slows down. Under the influence of intermolecular forces such as hydrogen bond and elastic contraction force caused by the restriction of the hard segment in the molecular chain, the internal stress is generated in the region where crystallization has occurred. The position and shape remain unchanged, the shape is fixed, the deformation recovery energy continues to be stored, and finally, the equilibrium is reached. Due to the weak intermolecular force and no hard segment fixation, the amorphous region without crystallization continues to produce chain slip, increasing deformation speed. After the external force is removed, the shape of the sample is fixed. In the final stage, the temperature rises rapidly from low temperature to the melting temperature of the sample. The recovery energy and shape fixed amount of the sample stored by hydrogen bond are released in a short time, and the shape begins to recover rapidly and finally reaches equilibrium. It is worth noting that HPE-0 with the highest PEMTC content has the least obvious differentiation in the third and fourth stages. We speculate that the high crystallinity of HPE-0 due to the short molecular chain, a high proportion of hard segment stationary phase, and uniform hydrogen bond aggregation leads to the low-temperature edge reaction of HPE-0. HPE-0 has a low degree of chain extension, a high degree of branching, and a short molecular chain, and its R_f value is significantly lower than other samples. It can be seen from the figure that except for HPE-0 samples, RF and RR of other samples will increase with the increase of PEMTC content. When the temperature is close to the melting temperature, the deformation can recover rapidly within 5 min. Therefore, the introduction of PEMTC can greatly improve the cross-linking strength, hydrogen bonding degree, shape recovery ability, mechanical properties, and heat resistance of HPE materials.

Conclusions

A series of polyurethane elastomers with shape memory function were synthesized by using PEMTC as a chain extender, and small molecule diol PEMTC was synthesized from IPDI, HEMAA, and TMP. It was found that the addition of PEMTC in the synthesis process of polyurethane can make the cross-linking between polyurethane molecules after 270 nm UV irradiation, a large number of functional groups which can form hydrogen bonds gather, the degree of hydrogen bond polymerization is improved, and the mechanical properties and shape memory function of the elastomer material is enhanced.

Conflicts of interest

There are no conflicts to declare.

Statement of contributions

Muqun Wang: Conceptualization, Data curation, Formal Analysis, Validation, Writing-original draft, Writing-review&editing. Shaofeng Liang: Writing-review&editing, Software. Wei Gao: Funding acquisition, Resources, Supervision. Yuxuan Qin: Writing-review&editing, Validation

Acknowledgements

Notes and references

1. H. R. Jarrah, A. Zolfagharian, R. Hedayati, A. Serjouei and M. Bodaghi, *Actuators*, 2021, **10**(3), 46.
2. J. Kim, S.-Y. Jeon, S. Hong, Y. An, H. Park and W.-R. Yu, *Smart Materials and Structures*, 2021, **30**, 3.
3. E. D'Elia, H. S. Ahmed, E. Feilden and E. Saiz, *Applied Materials Today*, 2019, **15**, 185-191.
4. Z.-X. Yang, X. Liu, Y. Shao, B. Yin and M.-B. Yang, *Polymer Composites*, 2019, **40**, E1353-E1363.
5. Y. Bai, Y. Chen, Q. Wang and T. Wang, *Journal of Materials Chemistry A*, 2014, **2**, 9169-9177.
6. A. C. Ferrari, F. Bonaccorso, V. Fal'ko, K. S. Novoselov, S. Roche, P. Boggild, S. Borini, F. H. L. Koppens, V. Palermo, N. Pugno, J. A. Garrido, R. Sordan, A. Bianco, L. Ballerini, M. Prato, E. Lidorikis, J. Kivioja, C. Marinelli, T. Ryhaenen, A. Morpurgo, J. N. Coleman, V. Nicolosi, L. Colombo, A. Fert, M. Garcia-Hernandez, A. Bachtold, G. F. Schneider, F. Guinea, C. Dekker, M. Barbone, Z. Sun, C. Galotit, A. N. Grigorenko, G. Konstantatos, A. Kis, M. Katsnelson, L. Vandersypen, A. Loiseau, V. Morandi, D. Neumaier, E. Treossi, V. Pellegrini, M. Polini, A. Tredicucci, G. M. Williams, B. H. Hong, J.-H. Ahn, J. M. Kim, H. Zirath, B. J. van Wees, H. van der Zant, L. Occhipinti, A. Di Matteo, I. A. Kinloch, T. Seyller, E. Quesnel, X. Feng, K. Teo, N. Rupesinghe, P. Hakonen, S. R. T. Neil, Q. Tannock, T. Loefwander and J. Kinaret, *Nanoscale*, 2015, **7**, 4598-4810.
7. L. Hu, R. Zhang and Q. Chen, *Nanoscale*, 2014, **6**, 14064-14105.
8. G. Yang, X. Liu, A. I. Y. Tok and V. Lipik, *Polymer Chemistry*, 2017, **8**, 3833-3840.
9. Q. Zhao, C. Li, H. C. Shum and X. Du, *Lab on a Chip*, 2020, **20**, 4321-4341.

10. Y. Bai, J. Zhang, D. Wen, B. Yuan, P. Gong, J. Liu and X. Chen, *Journal of Materials Chemistry A*, 2019, **7**, 20723-20732.
11. X. Zhang, Q. Zhou, H. Liu and H. Liu, *Soft Matter*, 2014, **10**, 3748-3754.
12. Z. H. Shen, K. Y. Liu, Z. Zhou and Q. F. Li, *J. Mat. Chem. B*, 2021, **9**, 992-1001.
13. W. Zhang and C. Gao, *Journal of Materials Chemistry A*, 2017, **5**, 16059-16104.
14. J. Kucinska-Lipka, I. Gubanska, H. Janik, M. Pokrywczynska and T. Drewa, *React Funct Polym*, 2015, **97**, 105-115.
15. J. Kucinska-Lipka, I. Gubanska, A. Lewandowska, A. Terebieniec, A. Przybytek and H. Cieslinski, *Polym Bull*, 2019, **76**, 2725-2742.
16. W. H. Fan, Y. Jin and L. J. Shi, *Polym Chem-Uk*, 2020, **11**, 5463-5474.
17. S. W. Yang, X. S. Du, S. Deng, J. H. Qiu, Z. L. Du, X. Cheng and H. B. Wang, *Chem Eng J*, 2020, **398**, 125654.
18. S. Chen, *Aatcc J Res*, 2021, **8**, 38-47.
19. N. K. Memis and S. Kaplan, *Tekstil ve Muhendis*, 2018, **25**, 9-21.
20. N. Abbasnezhad, N. Zirak, M. Shirinbayan, S. Kouidri, E. Salahinejad, A. Tcharkhtchi and F. Bakir, *J Appl Polym Sci*, 2021, **138**(12), 50083.
21. H. Yin, B. H. Du, Y. Chen, N. J. Song, Z. Li, J. H. Li, F. Luo and H. Tan, *J Biomat Sci-Polym E*, 2020, **31**, 2220-2237.
22. D. I. Arun, K. S. S. Kumar, B. S. Kumar, P. Chakravarthy, M. Dona and B. Santhosh, *Mater Sci Tech-Lond*, 2019, **35**, 596-605.
23. Y. Shi, G. Fang, Z. Cao, F. Shi, Q. Zhao, Z. Fang and T. Xie, *Chem Eng J*, 2021, 426, 131306.
24. H. G. Gui, G. W. Guan, T. Zhang and Q. P. Guo, *Chem Eng J*, 2019, **365**, 369-377.
25. S. Zhao, K. G. Battiston and J. P. Santerre, *Acs Biomater Sci Eng*, 2020, **6**, 4433-4445.
26. H. W. Cao, F. X. Y. Qi, R. W. Liu, F. T. Wang, C. X. Zhang, X. N. Zhang, Y. Y. Chai and L. L. Zhai, *Rsc Adv*, 2017, **7**, 11244-11252.
27. J. C. Dong, B. Y. Liu, H. N. Ding, J. B. Shi, N. Liu, B. Dai and I. Kim, *Polym Chem-Uk*, 2020, **11**, 7524-7532.
28. Y. Eom, S. M. Kim, M. Lee, H. Jeon, J. Park, E. S. Lee, S. Y. Hwang, J. Park and D. X. Oh, *Nat Commun*, 2021, **12**, 621.
29. Z. C. Feng, B. Zhu, H. X. Yao, M. J. Wang and X. H. Yang, *Acta Polym Sin*, 2012, 318-325.
30. S. Velankar and S. L. Cooper, *Macromolecules*, 1998, **31**, 9181-9192.
31. S. Velankar and S. L. Cooper, *Macromolecules*, 2000, **33**, 382-394.
32. Y. Han, H. Wang, X. Jiao and D. Chen, *Journal of Applied Polymer Science*, 2020, **137**(39), 49158.
33. J. Uchida, M. Yoshio and T. Kato, *Chemical Science*, 2021, **12**, 17.
34. H. Abushammala, *Polymers*, 2019, **11**(7), 1164.
35. Y.-S. Han, K.-Y. Jung, Y.-J. Kim, K. K. Baeck, G. M. Lee and S. W. Lee, *New Journal of Chemistry*, 2019, **43**, 15614-15625.
36. J. Ban, L. Mu, J. Yang, S. Chen and H. Zhuo, *Journal of Materials Chemistry A*, 2017, **5**, 14514-14518.
37. A. Nissenbaum, I. Greenfeld and H. D. Wagner, *Polymer*, 2020, **190**, 122226.
38. M. Hui, L. Yu-Cun, C. Tao, H. Tuo-Ping, G. Jia-Hu, Y. Yan-Wu, Y. Jun-Ming, W. Jian-Hua, Q. Ning and Z. J. e.-P. Liang, 2017, **17**(1), 89-94.
39. W. Sun, Y. Xin and Z. J. P. B. Xi, 2012, **69**, 621-633.
40. J. Wu, L. H. Cai and D. A. J. A. M. Weitz, 2017, 1702616.
41. X. Chen, Q. Zhong, C. Cui, L. Ma and Y. Zhang, *ACS Applied Materials&Interfaces*, 2020, **12**, 30847-30855.
42. H. Hu and D. Dorset, *Macromolecules*, 1990, **23**, 2, 623-633.
43. S. Nojima, K. Hashizume, A. Rohadi and S. Sasaki, *Polymer*, 1997, **38**, 2711-2718.

Fig. 1 The diagram of synthesis of PEMTC.

Fig. 2 The diagram of synthesis of HPEs.

28 Fig. 3 FT-IR spectra of PEITC and its reactants (a); FT-IR spectra of PEMTC and its reactants (b); ^1H NMR spectrum of PEMTC.
29 Table 1. The assignments of ^1H -NMR spectrum of PEMTC.

Fig. 4 FT-IR spectra of HPE-6 before and after UV irradiation (a); FT-IR spectra of HPEs (b); FTIR spectra of HPE-6 film before UV

Fig. 5 Stress-strain curves of HPEs before UV irradiation (a); stress-strain curves of HPEs after UV irradiation (b); the elongation at break of HPEs before and after UV irradiation (c); the break strength of HPEs before and after UV irradiation (d); the elasticity modulus of HPEs before and after UV irradiation (e); the toughness of HPEs before and after UV irradiation (f).

1
2
3
4
5
6
7
8
9
10
11
12
13
14
15
16
17
18
19
20
21
22
23
24
25
26
27
28
29
30
31
32
33
34
35
36
37
38
39
40
41
42
43
44
45
46
47
48
49
50
51
52
53
54
55
56
57
58
59
60

Fig. 6. Wide-angle X-ray diffraction (WAXD) curves of HPEs

Fig. 7 Schematic diagram of polymerization principle.

Fig. 8 TEM images of samples HPE-0 (a), HPE-3 (b), HPE-6 (c), HPE- 8(d), and HPE- 9(e); the scale bar is 50 nm.

1
2
3
4
5
6
7
8
9
10
11
12
13
14
15
16
17
18
19
20
21
22
23
24
25
26
27
28
29
30
31
32
33
34
35
36
37
38
39
40
41
42
43
44
45
46
47
48
49
50
51
52
53
54
55
56
57
58
59
60

Fig. 9 TGA curves of HPEs (a); DTG curves of HPEs (b); DTA curves of HPEs (b); Melting temperature and decomposition temperature of HPEs.

Fig. 10 Shape memory behaviour of HPE-6 at 60 °C.

1
2
3
4
5
6
7
8
9
10
11
12
13
14
15
16
17
18
19
20
21
22
23
24
25
26
27
28
29
30
31
32
33
34
35
36
37
38
39
40
41
42
43
44
45
46
47
48
49
50
51
52
53
54
55
56
57
58
59
60

Fig. 11 Stress-strain curve of HPEs at 60 °C(a); Shape memory DMA curves of HPE-0(b), HPE-3(c), HPE-6(d), HPE-8(e) and HPE-9(f).

1
2
3
4
5
6
7
8
9
10
11
12
13
14
15
16
17
18
19
20
21
22
23
24
25
26
27
28
29
30
31
32
33
34
35
36
37
38
39
40
41
42
43
44
45
46
47
48
49
50
51
52
53
54
55
56
57
58
59
60

Table 1. The assignments of ^1H -NMR spectrum of PEMTC.

Position of proton	Chemical shift (ppm)	Theoretical number of proton	Integral results of the peaks
a	5.80~6.57	3	2.72
b	4.21~4.53	4	3.92
j	4.12	2	2.03
c+k	3.58~3.86	5	4.97
l	3.5	2	2.29
i	2.91	2	1.97
d+h	1.15~1.44, 1.56~1.83	4	3.97
e+f+g+h+m+n	0.72~1.15	18	16.72

Appendix B

In this manuscript, the authors presented PEMTC-incorporated polyurethane for inducing crosslinking between the polyurethane chains upon UV irradiation. They synthesized a series of polyurethane elastomers using different ratios of chain extenders: TMP and PER (branched-chain extenders) and BDO and PEG (straight-chain extenders). The chemical structures of synthesized materials were thoroughly investigated using FT-IR and NMR. To understand the effect of UV-driven crosslinking and hydrogen bonding on their mechanical properties, stress-strain curves, elongation at break, elasticity modulus, and toughness of the polymers before and after UV irradiation were analyzed. The shape memory behavior of the polymers is well presented as well. Finally, the authors suggested the tensile process mechanism dividing into 5 different stages through the analysis of stress-strain curves of the samples at different temperatures. I believe the paper can be a good addition to the field and therefore I recommend publication of the manuscript in Royal Society Open Science after addressing the following comments.

1. While HPE-0 used 0.01 mol of PEMTC without chain extender or additional IPDI to compare with other samples with chain extenders, I believe a sample with 0.005 mol of PEMTC and without chain extender should be included and compared with other samples because most of the samples only include 0.005 mol of PEMTC.
2. Even though the authors prepared a series of HPE samples, from HPE-0 to HPE-9 as shown in Table S1, not all the material properties (e.g. mechanical property, x-ray data, and TEM images) of these samples are presented but they only showed those of HPE-0, -3, -6, -8, and -9. Please provide all the data for understanding the property-composition relationship.
3. In Figure 8, while the authors claimed that the micro cross-linking of the sample can be seen intuitively, I'm not sure what can be claimed based on the TEM data. I believe the TEM data should be further analyzed in a more quantitative manner to avoid any misleading for the readers.
4. The unit of temperature in Figure 9 must be written properly.
5. What is the reason that the authors presented the shape memory behavior of HPE-6 only? It would help to include and compare the shape memory behavior of other samples as well.

Appendix C

Dear editor :

Title: The effect of promoting hydrogen bond aggregation based on PEMTC on the mechanical properties and shape memory function of polyurethane elastomers

ID: RSOS-211393.R1

We have completed the revision of the article, and re-uploaded the relevant documents and responses to the reviewers. Please check and deal with it.

Warm regards,

Muqun Wang
School of Resources, Environment and Materials, Guangxi University

Reviewer: 1

Comments to the Author(s)

The author synthesized a polyurethane structure modified with carbon-carbon double bonds, and studied the effect of hydrogen bonding on the shape memory effect by means of ultraviolet light-initiated polymerization. However, many statements in the article lack data support, such as how to use XRD to know the change of aggregation degree? How to distinguish the effect of hydrogen bond shape memory rather than the degree of crosslinking? How to prove that hydrogen bonds can increase or decrease the bond length of -C=O and -NH-? No DMA data is used to characterize shape memory performance. And there are many grammatical and input errors. Other issues are highlighted in the attachment.

Author's response: We have revised the text one by one with reference to the reviewers' comments. The specific modification is shown in the attachment Modify content.docx. The XRD image analysis of HPEs and PCL in Figure 6 of the main text can indicate the degree of microphase separation of the sample; figure 4 in the main text and Figure S3 in the supplementary material show the quantitative analysis of hydrogen bond content in HPEs samples; figure 11 and pages 10 to 11 of the main text are the qualitative analysis of the shape memory characteristics of HPE samples.

Reviewer: 2

Comments to the Author(s)

The manuscript reported a polyurethane elastomers with shape memory function, but the formation of reported elastomers are too complicated and writing should be improved. HRMS of PEITC and PEMTC should be provided. The results of small-angle X-ray scattering should also be provided.

Author's response: The main content of this article is to introduce a synthetic method to improve the shape memory performance of polyurethane materials. This method achieves the purpose of reducing the amount of raw materials by optimizing the synthesis process. PEITC and PEMTC are only intermediate products in the synthesis process of HPEs, not the core content of the article. The NMR test results of PEMTC are shown in Figure 3 and Table 1 of the main text.

Reviewer: 3

In this manuscript, the authors presented PEMTC-incorporated polyurethane for inducing crosslinking

between the polyurethane chains upon UV irradiation. They synthesized a series of polyurethane elastomers using different ratios of chain extenders: TMP and PER (branched-chain extenders) and BDO and PEG (straight-chain extenders). The chemical structures of synthesized materials were thoroughly investigated using FT-IR and NMR. To understand the effect of UV-driven crosslinking and hydrogen bonding on their mechanical properties, stress-strain curves, elongation at break, elasticity modulus, and toughness of the polymers before and after UV irradiation were analyzed. The shape memory behavior of the polymers is well presented as well. Finally, the authors suggested the tensile process mechanism dividing into 5 different stages through the analysis of stress-strain curves of the samples at different temperatures. I believe the paper can be a good addition to the field and therefore I recommend publication of the manuscript in Royal Society Open Science after addressing the following comments.

1. While HPE-0 used 0.01 mol of PEMTC without chain extender or additional IPDI to compare with other samples with chain extenders, I believe a sample with 0.005 mol of PEMTC and without chain extender should be included and compared with other samples because most of the samples only include 0.005 mol of PEMTC.

Author's response: In order for the chain extension reaction to proceed normally, it should be ensured that the molar ratio of -NCO to -OH contained in the system is the same. According to the reaction design during the experiment, in order to avoid the uncontrollable composition of the hard segment in the product, we did not consider the self-polymerization caused by the reaction of -NCO with air moisture and CO₂. After adding 0.005mol PEMTC, only 0.01mol PCL participated in the reaction. After the sample is purified, unreacted PCL will be removed, and the result will be the same as HPE-0.

2. Even though the authors prepared a series of HPE samples, from HPE-0 to HPE-9 as shown in Table S1, not all the material properties (e.g. mechanical property, x-ray data, and TEM images) of these samples are presented but they only showed those of HPE-0, -3, -6, -8, and -9. Please provide all the data for understanding the property-composition relationship.

Author's response: We explained why we only selected five groups of samples of HPE-0, HPE-3, HPE-6, HPE-8 and HPE-9 as controls at “*We can first synthesize linear macromolecular samples HPE-1, ... By designing different proportions, samples HPE-7, HPE-8, and HPE-9, and HPE-10 are obtained.*” on page 5 of the main text. Since the HPEs samples were finally synthesized through three stages, we only selected the samples with the highest contrast in each stage for comparison. The mechanical performance parameters of all samples in the second stage are similar and have no reference value, so we did not analyze them.

3. In Figure 8, while the authors claimed that the micro cross-linking of the sample can be seen intuitively, I'm not sure what can be claimed based on the TEM data. I believe the TEM data should be further analyzed in a more quantitative manner to avoid any misleading for the readers.

Author's response: We have analyzed and written this part of the content at “*Fig. 7 is a schematic diagram of the polymerization principle of HPEs... In Fig. 8, through TEM images, we can see the micro cross-linking of the sample more intuitively. The aggregation degree of HPES micelles can be improved by increasing the amount of PEMTC. The white dotted circle indicates the typical island structure.*” on page 8 of the main text. TEM is only used as a way to observe the obvious degree of the structure of the sample ring island, not as the main reference.

4. The unit of temperature in Figure 9 must be written properly.

Author's response: We have optimized Figure 9.

5. What is the reason that the authors presented the shape memory behavior of HPE-6 only? It would help to include and compare the shape memory behavior of other samples as well.

Author's response: The display of the shape memory image of HPE-6 shows that the synthesized HPEs have a good shape memory function. We have added video demonstrations of the shape memory behavior of the remaining samples in the supplementary materials.